# Nerve growth factor transfer from cardiomyocytes to innervating sympathetic neurons activates TrkA receptors at the neuro-cardiac junction

Lolita Dokshokova[1,2,3] (ID), Mauro Franzoso[1], Anna Di Bona[4], Nicola Moro[1] (ID), Jose Luis Sanchez Alonso[3], Valentina Prando[1] (ID), Michele Sandre[1] (ID), Cristina Basso[4], Giuseppe Faggian[2] (ID), Hugues Abriel[5] (ID), Oriano Marin[1], Julia Gorelik[3], Tania Zaglia[1] and Marco Mongillo[1,6] (ID)

[1]*Department of Biomedical Sciences, University of Padova, Padova, Italy*

[2]*Division of Cardiac Surgery, University of Verona, Verona, Italy*

[3]*National Heart and Lung Institute, London, UK*

[4]*Department of Cardiac, Thoracic, Vascular Sciences and Public Health, University of Padova, Padova, Italy*

[5]*Institute of Biochemistry and Molecular Medicine, University of Bern, Bern, Switzerland*

[6]*CNR Institute of Neuroscience, Padova, Italy*

Edited by: Bjorn Knollmann & Jian Shi

Linked articles: This article is highlighted in a Perspective article by Salamon & Mahmoud. To read this article, visit https://doi.org/10.1113/JP283173.

The peer review history is available in the supporting information section of this article (https://doi.org/10.1113/JP282828#support-information-section).

*The Journal of Physiology*

L. Dokshokova and M. Franzoso contributed equally to this work.

**Abstract** Sympathetic neurons densely innervate the myocardium with non-random topology and establish structured contacts (i.e. neuro-cardiac junctions, NCJ) with cardiomyocytes, allowing synaptic intercellular communication. Establishment of heart innervation is regulated by molecular mediators released by myocardial cells. The mechanisms underlying maintenance of cardiac innervation in the fully developed heart, are, however, less clear. Notably, several cardiac diseases, primarily affecting cardiomyocytes, are associated with sympathetic denervation, supporting the hypothesis that retrograde 'cardiomyocyte-to-sympathetic neuron' communication is essential for heart cellular homeostasis. We aimed to determine whether cardiomyocytes provide nerve growth factor (NGF) to sympathetic neurons, and the role of the NCJ in supporting such retrograde neurotrophic signalling. Immunofluorescence on murine and human heart slices shows that NGF and its receptor, tropomyosin-receptor-kinase-A, accumulate, respectively, in the pre- and post-junctional sides of the NCJ. Confocal immunofluorescence, scanning ion conductance microscopy and molecular analyses, in co-cultures, demonstrate that cardiomyocytes feed NGF to sympathetic neurons, and that this mechanism requires a stable intercellular contact at the NCJ. Consistently, cardiac fibroblasts, devoid of NCJ, are unable to sustain SN viability. ELISA assay and competition binding experiments suggest that this depends on the NCJ being an insulated microenvironment, characterized by high [NGF]. In further support, real-time imaging of tropomyosin-receptor-kinase-A vesicle movements demonstrate that efficiency of neurotrophic signalling parallels the maturation of such structured intercellular contacts. Altogether, our results demonstrate the mechanisms which link sympathetic neuron survival to neurotrophin release by directly innervated cardiomyocytes, conceptualizing sympathetic neurons as *cardiomyocyte-driven* heart drivers.

(Received 11 January 2022; accepted after revision 28 March 2022; first published online 12 April 2022)

**Corresponding author** Marco Mongillo and Tania Zaglia: Department of Biomedical Sciences, University of Padova, Via Ugo Bassi 58/B, 35121 Padova, Italy. Email: marco.mongillo@unipd.it and tania.zaglia@unipd.it.

**Abstract figure legend** Sympathetic neuron (SN, green) varicosities establish synaptic contacts with target cardiomyocytes (CMs, pink), which we previously called neuro-cardiac junctions (NCJ, Prando et al., 2018). At NCJs, CMs selectively release NGF, which by activating TrkA signalling, is key to sustaining neuronal survival.

### Key points

- CMs are the cell source of nerve growth factor (NGF), required to sustain innervating cardiac SNs;
- NCJ is the place of the intimate liaison, between SNs and CMs, allowing on the one hand neurons to peremptorily control CM activity, and on the other, CMs to adequately sustain the contacting, ever-changing, neuronal actuators;
- alterations in NCJ integrity may compromise the efficiency of 'CM-to-SN' signalling, thus representing a potentially novel mechanism of sympathetic denervation in cardiac diseases.

**Lolita Dokshokova, MSc, PhD,** is a physicist, who recently completed her PhD in Cardiovascular Sciences, at the University of Verona. Lolita spent part of her PhD in Padova, in the Zaglia and Mongillo laboratories, and part in London, in Gorelik's lab. The synergy of skills in physics, with those acquired in biology, allowed Lolita to implement *in vitro* co-culture systems and study intercellular signalling with advanced microscopy techniques. **Mauro Franzoso, MSc, PhD,** is a biotechnologist, who received his PhD in Biomedical Sciences, at the University of Padova. Mauro dedicated himself to *in vitro* and *ex vivo* studies aimed at dissecting the mechanisms regulating heart sympathetic innervation, with a focus on the retrograde, neurotrophin-mediated, 'cardiomyocyte-to-sympathetic neuron' communication, in physiology and pathology. The research pursued by Drs Dokshokova and Franzoso has recently been awarded *best poster presentation* at the European section of the International Society for Heart Research Congress (ISHR 2021), held in Turin.

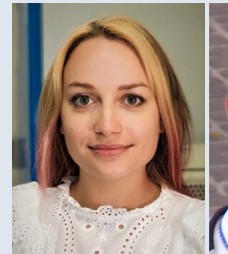
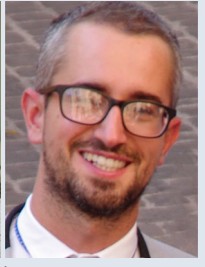

## Introduction

Sympathetic neurons (SNs) densely innervate the heart of mammals, including humans (Kawano et al., 2003; Wichter et al., 2000) with a well-defined topology, which is finely regulated by the myocardium itself (Franzoso et al., 2016; Ieda et al., 2004; Ieda et al., 2007; Zaglia & Mongillo, 2017). The precise species-specific geometry of myocardial sympathetic innervation ensures optimal electrical and contractile performance, and allows continuous adaptation of heart function through acute (i.e. chronotropic, inotropic and lusitropic) and chronic (i.e. regulation of gene expression) effects of neuronal inputs on target cardiomyocytes (CMs) (Franzoso et al., 2016; Larsen et al., 2016; Pianca et al., 2019; Prando et al., 2018; Shan et al., 2010; Zaglia et al., 2013). Dysfunction in the neurogenic control of cardiac activity features in several cardiovascular diseases, including myocardial hypertrophy, ischaemia/infarction (MI) and heart failure, all conditions characterized by increased arrhythmic incidence, associated with either enhanced sympathetic drive or reduced SN transmission (Gardner et al., 2016; Hasan et al., 2006; Herring et al., 2019; Kaye et al., 2000; Kimura et al., 2007; Miyauchi et al., 2003; Schäfers et al., 1998; Zhou et al., 2004). While the effects of cardiac hyperinnervation, heightening CM neuronal stimulation, have been the subject of intense research, and are now well-defined (Cao, Chen et al., 2000; Cao, Fishbein et al., 2000; Franzoso et al., 2016; Miyauchi et al., 2003), the consequence of cardiac SN (cSN) degeneration is only recently being appreciated. On this trail, while cardiac neurons may primarily be affected in several neuro-degenerative disorders (e.g. Parkinson's, Huntington's disease) (Kiriazis et al., 2012; Kobal et al., 2004; Orimo et al., 2007), in other cases, SN degeneration follows CM defect (e.g. familial hypertrophic cardiomyopathies) (Li et al., 2000; Schäfers et al., 1998; Terai et al., 2003). Such observations guided us to investigate the mechanisms of retrograde 'CM-to-SN' communication, and its role in the homeostasis of cardiac sympathetic innervation.

Recently, in line with previous evidence (Shcherbakova et al., 2007), we demonstrated that each cSN varicosity contacts target CMs at the neuro-cardiac junction (NCJ), an intercellular contact structure which, by analogy to the well-known neuro-muscular junction, allows neurons to interact intimately with CMs, enabling cardiac activity to be controlled with high precision and efficiency (Pianca et al., 2019; Prando et al., 2018). It remains elusive, however, whether such direct 'cell-to-cell' coupling is also involved in reverse CM-to-SN signalling, which would thus identify the NCJ as a hub of bidirectional intercellular communication in the heart.

It is well known that CMs synthesize and release neurotrophic and neuro-repellent factors, together tailoring the establishment and topology of the cardiac sympathetic network during development (Glebova & Ginty, 2004; Habecker et al., 2008; Habecker et al., 2016; Ieda et al., 2004; Ieda et al., 2007; Lockhart et al., 1997; Lorentz et al., 2010). In addition, SN viability depends on the continuous neurotrophin supply (mainly nerve growth factor (NGF) in the heart), typically provided by the target organ (Habecker et al., 2008; Habecker et al., 2016; Heumann et al., 1984; Lockhart et al., 1997; Shelton & Reichardt, 1984; Zweifel et al., 2005). Consistently, post-ischaemic sympathetic denervation has been attributed to CM damage, affecting neurotrophin-dependent SN sustainment.

Thus, although the general concept that sympathetic innervation is regulated by the cardiac muscle is acknowledged, the detailed physiology of 'CM-to-neuron' neurotrophin exchange has been poorly explored. To address this point, we combined the analysis of murine and human heart with *in vitro* assays in SN/CM co-cultures. The morphology and biophysics of the NCJ were characterized using confocal microscopy, scanning ion conductance microscopy (SICM), morphometric analyses, live imaging of NGF-receptor (NGF-R) trafficking, gene silencing and pharmacological tests. Our results show that CMs directly exchange NGF with the innervating neurons, and that the NCJ is the functional unit where muscle cells nourish contacting neurons, supporting the hypothesis that integrity of the cardiac sympathetic network relies on properly-functioning CM-to-SN communication

## Methods

We declare that all investigators involved in the study understand the ethical principles under which the journal operates and that the work complies with the animal ethics checklist of the journal (Grundy, 2015).

### Human heart sample processing and immunofluorescence

We analysed heart samples from three male subjects (age: 45 ± 8 years) who had died from extra-cardiac causes (accidents), acquired during routine post-mortem clinical investigations, and archived at the Institute of Pathological Anatomy of the University of Padova. Samples were anonymous to the investigators and used in accordance with the 'Recommendation (CM/Rec(2016)6) of the Committee of Ministers to Member States on research on biological materials of human origin', released by the Council of Europe, as received by the Italian National Council of Bioethics. Samples were analysed using protocols previously described in Zaglia et al. (2016). The primary and secondary antibodies, and chemicals, used in this study are listed in Tables 1–3.

**Table 1. Primary antibodies used in this study**

| Target | Company | Host | Dilution |
|---|---|---|---|
| α-actinin | Sigma-Aldrich | mouse | 1:200 |
| cardiac troponin I | (Saggin et al., 1989) | mouse | 1:200 |
| nerve growth factor | Abcam | rabbit | 1:100 |
| SNAP25 | Abcam | mouse | 1:200 |
| tyrosine hydroxylase | Millipore | rabbit | 1:400 |
| tyrosine hydroxylase | Millipore | sheep | 1:100 |
| TrkA | Alomone | rabbit | 1:200 |
| vimentin | Sigma-Aldrich | mouse | 1:200 |

**Table 2. Secondary antibodies used in this study**

| Antibody | Company | Host | Dilution |
|---|---|---|---|
| Anti-mouse Alexa Fluor-647 | Invitrogen | rabbit | 1:300 |
| Anti-mouse Cy3 | Jackson | goat | 1:200 |
| Anti-rabbit Alexa Fluor 488 | Jackson | goat | 1:200 |
| Anti-rabbit Alexa Fluor 647 | Jackson | donkey | 1:200 |
| Anti-rabbit Cy3 | Jackson | goat | 1:200 |
| Anti-sheep TRITC | Jackson | donkey | 1:100 |

**Table 3. Chemicals used in immunofluorescence experiments**

| Name | Company | Dilution |
|---|---|---|
| Alexa Fluor 633-phalloidin | Invitrogen | 1:200 |
| DAPI | Invitrogen | 1:5000 |

### Ethical approval

All experimental procedures in murine models were approved by the Ministry of Health (Ufficio VI), in compliance with the Animal Welfare Legislation (VIMM C-53 and C-54). All procedures were performed by personnel with documented formal training and previous experience in experimental animal handling and care. All procedures were refined prior to initiating the study, and the number of animals was calculated to use the least number needed to achieve statistical significance, according to sample power calculations.

### Origin and source of animals

In this study, we used P1–P3 and adult (3 months old.) Sprague-Dawley male rats (Harlan, Milan, Italy). Animals were maintained in individually ventilated cages in an authorized animal facility (authorization number 175/2002A) under a 12:12 h light/dark cycle at a controlled temperature and had access to water and food available *ad libitum*. Rats were killed by cervical dislocation (in accordance with Annex IV of European Directive 2010/63/EU). In adult rats, sedation with 3% isoflurane (v:v in $O_2$) was performed before cervical dislocation.

### Immunofluorescence analysis of rodent hearts

Hearts were harvested from adult rats, fixed in 1% paraformaldehyde (PFA) (w:v in PBS; Sigma Aldrich), and processed as described in Zaglia et al. (2013). Cryosections 10 μm thick were obtained using a cryostat (Leica 1860) and processed for immunofluorescence (IF), as previously described (Zaglia et al., 2013). The primary and secondary antibodies, as well as chemicals used in this study are listed in Tables 1–3.

### Analysis of SNAP25 and nerve growth factor distribution at the neuro-cardiac contacts

Analysis was performed on confocal z-series images post-processed using ImageJ (Wayne Rasband, Bethesda, MD, USA). Three-dimensional images were rendered and fluorescence intensity and displacement were measured along parallel lines manually drawn in correspondence with, or away from, the NCJ, identified by morphology.

### Establishment of sympathetic neuron/cardiomyocyte co-cultures

Co-cultures between SNs isolated form the superior cervical and stellate ganglia, and CMs from P1–P3 neonatal rats, were set up as described in Pianca et al. (2019) and Prando et al. (2018). We here analysed 2, 4, 7, 10 and 14 day co-cultures. In a subset of experiments (i.e. assessment of TrkA movements), co-cultures were established between rat neonatal CMs (P1–P3) and PC12-derived SNs, as described in Prando et al. (2018).

### Establishment of sympathetic neuron/cardiac fibroblasts co-cultures

Cardiac fibroblasts (CFs) were obtained by plastic adhesion-based separation during CM preparation. Cells were expanded, seeded in laminin-coated coverslips at a density of 100 cells/mm². The SN:CF ratio was 1:25. Cells were maintained in the same culture conditions and analysed at the same time points as SN/CM cultures (see above).

### Scanning ion conductance microscopy

SICM is a contactless imaging method in which the surface of a cellular sample is scanned by an electrolyte solution-loaded nanopipette, continuously measuring the resistance established between the cell and pipette,

which depends on their reciprocal distance (Hansma et al., 1989). All topographical images were recorded using the SICM hopping mode (Novak et al., 2009). The tip size of the nanopipette was ~100 nm (80–100 MΩ), pulled from a borosilicate glass capillary (IntraCel BF100-50-7.5) with a laser puller (ItraCel, Sutter Instrument Co, P-2000). Cells were bathed in 'extracellular' solution, while the pipette was filled with 'intracellular' solution. The pipette manipulator operates vertical movements along the Z direction, towards the cell surface, and by recording the pipette coordinates when a predetermined current resistance is detected, a topographic map of the scanned surface can be obtained using the software routine SICMView. The three-dimensional topological characteristics of the scanned object, such as volume, surface area, surface contact and height were calculated by a SICM image viewer (Novak et al., 2009).

### *In vitro* immunofluorescence analysis

Cells were fixed with 3.7% formaldehyde (Sigma) at 4°C for 30 min, permeabilized with 0.1% Triton (v:v in PBS) (Sigma Aldrich) for 5 min and incubated with the appropriate primary antibody for 2 h at 37°C. The primary and secondary antibodies, and chemicals, used in this study are listed in Tables 1–3.

### Evaluation of sympathetic neuron varicosity morphometry

To analyse the size of varicosities, regions of interest were manually drawn on tyrosine hydroxylase (TH)-positive enlargements along neuronal processes and quantitated on the maximal projection image obtained from a 10-image series along the *z*-axis and rendered using ImageJ. Enlargements were defined as axonal segments more than twice as large as the average axonal thickness in the same sample. The inter-varicosity distance was then measured with Image J, by calculating the distance along a line manually drawn between subsequent varicosities.

### *In vitro* transmission electron microscopy analysis

Transmission electron microscopy analysis was performed in SN/CM co-cultures, following the protocol described in Prando et al. (2018).

### Imaging of TrkA-DsRed2 vesicle movements in co-cultures

SN/CM co-cultures were established as described above. SNs were infected with an adenoviral vector encoding TrkA-DsRed2 (Vector BioLabs) at a multiplicity of

**Table 4. Oligoes used in RTqPCR experiments**

| Primer name | Sequence |
|---|---|
| Rat NGF Forward | 5′-TGACAGTGCTGGGCGAGGTGAA-3′ |
| Rat NGF Reverse | 5′-TCAATGCCCCGGCATCCACTCT-3′ |
| Rat NT3 Forward | 5′-CATAAGAGTCACCGAGGAGAGTACT-3′ |
| Rat NT3 Forward | 5′-ATGTCAATGGCTGAGGACTTGTC-3′ |
| Rat GAPDH Forward | 5′-AGGGCTGCCTTCTCTTGTGAC-3′ |
| Rat GAPDH Reverse | 5′-TGGGTACAGTCATACTGGAACATGTAG-3′ |
| Rat β actin Forward | 5′-CTGGCTCCTAGCACCATGAAGAT-3′ |
| Rat β actin Reverse | 5′-GGTGGACAGTGAGGCCAGGAT-3′ |

infection of 35. After 24 h, the virus was removed, and the medium freshly replaced, and imaging was performed at 48 h from infection. In detail, we compared TrkA dynamics in 4-day (early) *vs.* 14-day (mature) co-cultures. During imaging experiments, cells were maintained in tyrode solution (125 NaCl, 5 KCl, 1 $Na_3PO_4$, 1 $MgSO_4$, 5.5 glucose, 1.8 $CaCl_2$, 20 Hepes, in m M, pH 7.4). Cells were analysed in a culture dish incubator, at controlled temperature (37°C), atmosphere and pH. TrkA-DsRed2 vesicle movements were recorded by acquiring images every 30 s with a confocal microscope (Leica SP5), equipped with an oil immersion 1,3NA, 63X objective. Kymographs of the time-lapse images were obtained with the ImageJ plugin *kymograph*, as described in Jakobs et al. (2019).

### Cultured cardiomyocyte transfection

Cultured CMs were transfected using transfectin (Bio Rad), following the manufacturer's instructions. Cells were co-transfected with a plasmid-encoding green fluorescent protein (GFP) and small hairpin RNAs, including: shRNA NGF mRNA (encoding for the two transcript variants of the β polypeptide (XM_003749364.1, XM_227525.6) (Sigma-Aldrich)); SIC001 (Sigma-Aldrich) as control.

### RTqPCR analysis

The analysis was performed following the protocol described in Zaglia et al. (2014). The oligoes used in this study are listed in Table 4.

### Western blotting

This procedure was performed as described in Zaglia et al. (2014).

## ELISA assay

To estimate NGF concentration in the CM conditioned medium, we used the ChemiKine NGF sandwich ELISA Kit (Chemicon), following the manufacturer's instructions.

## Statistical analysis

Statistical analysis was performed using GraphPad Prism 8. Normality of data distribution was assessed with the Shapiro–Wilk test. An unpaired *t* test (for two groups) or one-way ANOVA, with Brown–Forsythe's and Welch's corrections (for three or more groups) were used for normally distributed data. A non-parametric Mann unpaired *t* test (for two groups) or one-way ANOVA (for three or more groups) were used for normally distributed data with equal variance. An unpaired *t* test with Welch's correction (for two groups) or Brown–Forsythe's and Welch's ANOVA (for three or more groups) were used for normally distributed data with unequal variance. The Mann–Whitney test (for two groups) or Kruskal–Wallis test (for three or more groups) were used for non-normally distributed data. Data distribution is represented by individual values, with means and error bars representing the 95% confidence interval. A *P* value <0.05 was considered statistically significant.

## Results

### In mammalian hearts, the neurotrophin-signalling elements preferentially accumulate at the neuro-cardiac interface

We have shown that, in mammalian hearts, structured interaction between catecholamine-releasing neuronal varicosities and CM membrane (i.e. NCJ) underpins localized and cell-directed *anterograde* neuro-cardiac communication (Prando et al., 2018). Here, we tested the hypothesis that *retrograde* 'CM-to-SN' neurotrophic signalling could also take place electively at the NCJ.

Confocal IF in ventricular cryosections of mammalian hearts showed that the vast majority of neuronal processes are sympathetic, and as such, markers of neuro-exocytosis (e.g. SNAP25) were used as *bona fide* indicators of cSNs (Prando et al., 2018). Co-IF in rat heart sections demonstrated that SNAP25 positive neuronal processes also express the high-affinity NGF receptor, TrkA, which was found in the enlargements identifiable as varicosities (Fig. 1*A*). Staining of the sections with an anti-NGF antibody showed immuno-reactive vesicles in CMs, which clustered in a roughly 2 $\mu$m deep submembrane space, and appeared to mirror the position of the pre-synaptic neuronal varicosities, in the majority (>60%) of neuro-cardiac interfaces analysed

(Fig. 1*B*). Consistently, sub-microscopic analysis of the immunostained sections revealed high intensity NGF puncta (in green) spread in the CM cytosol, accompanied by neurotrophin clusters aligned along the portion of the CM membrane directly contacted by the SN (Fig. 1*C*). This was in line with the results of morphometric image analysis aimed to trace pre-synaptic SNAP25 and post-synaptic NGF distribution, showing that the highest NGF fluorescence intensity, in the CM, corresponded to the peak of SNAP25 signal on the neuron (Fig. 1*D*). In addition, NGF was detected along the neuro-nal process, suggesting that CM-released neurotrophin could be sensed and locally internalized, via TrkA, by the contacting SN (Fig. 1*B*). Interestingly, the presence of NGF in CM-innervating neuronal processes and the preferential localization of the neurotrophin in the CM submembrane space, at neuro-cardiac contacts, were detected, and confirmed by morphometric analysis, in myocardial samples from post-mortem human hearts, proving the principle that the aspects described above hold true in the normal human heart (Fig. 2*A–C*).

Thus, the arrangement of NGF vesicles in CMs and of NGF-sensing receptors on SNs, suggests that exchange of neurotrophin between the two cell types may take place locally, and that the NCJ may thus be poised to sustain retrograde (myocyte-to-neuron) neurotrophic signalling.

### The neuro-cardiac junction matures in time while neurons become independent from exogenous NGF

To investigate retrograde neurotrophin-mediated communication at the single cell level, we used a pre-viously validated *in vitro* system, based on co-cultures between SNs and CMs (Pianca et al., 2019; Prando et al., 2018). Our data confirmed the common recognition (Greene, 1977; Mains & Patterson, 1973) that SNs, cultured *in vitro*, require the addition of NGF to the medium for survival and maturation (Fig. 3*A* and *B*). In fact, absence of NGF in the culture medium only allowed a negligible and unquantifiable amount of neurons to be detected in culture after 7 days. Remarkably, we soon noted, however, that when SNs were co-cultured with CMs, they developed normally, irrespective of exogenous NGF (Fig. 3*C–E*). We excluded the possibility that this resulted from the presence of NGF in the serum, as our co-culture medium was serum-free.

Based on our previous results (Prando et al., 2018), we compared (early) 7-day *vs.* (mature) 14-day co-cultures and, notably, we did not observe differences in SN density (number of cell soma/area (mm$^2$), 7 days, +NGF:17.05 $\pm$ 7.60 *vs.* -NGF:14.99 $\pm$ 11.61; 14 days, +NGF:10.42 $\pm$ 7.80 *vs.* 14 days, -NGF:9.39 $\pm$ 3.29, data expressed as means $\pm$ SD; *P* = ns) and in the morphology of neuro-nal processes (i.e. size of varicosities and interindividual

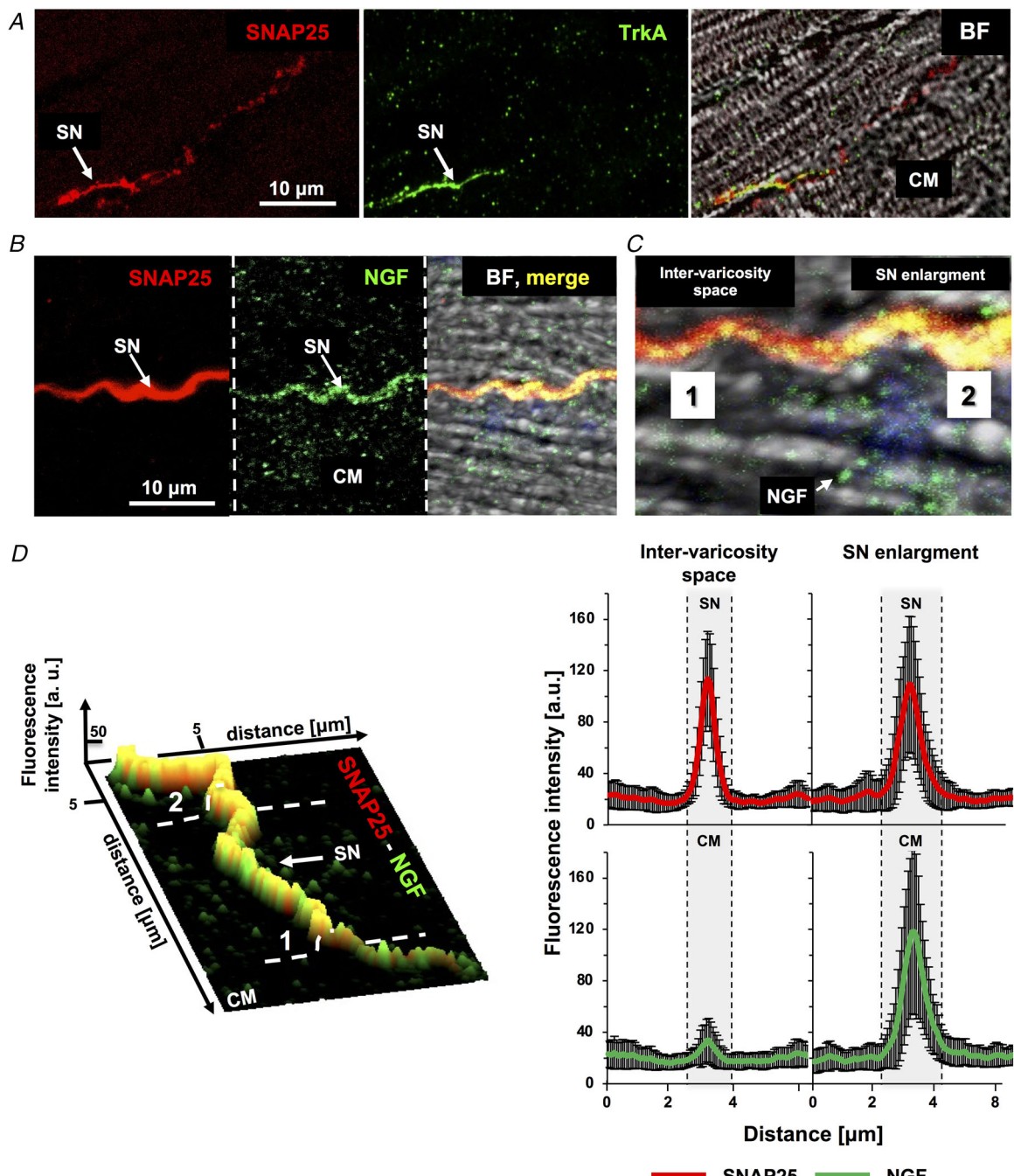

**Figure 1. The molecular players of nerve growth factor signalling predominantly localize at the neuro-cardiac junction, in rat hearts**

*A*, confocal immunofluorescence (IF) analysis of heart sections from adult rats, co-stained with antibodies against SNAP25 (left panel) and TrkA (middle panel). The right panel shows the bright field images (BF). SN, sympathetic neuron; CM, cardiomyocyte. The white arrow indicates the neuronal process. *B*, confocal IF analysis of adult rat heart sections, co-stained with antibodies against SNAP25 (left panel) and nerve growth factor (NGF) (middle panel). The right panel shows merged fluorescence and bright field images. The white arrow indicates the SN process. *C*, magnified image of panel *B*. *D*, surface rendering of the z-section series of the IF images shown in (*B*) and relative quantification of the fluorescence intensity of SNAP25 (red signal) and NGF (green signal) in correspondence with the neuronal varicosity (2) or the inter-varicosity space (1). Bars represent SD (*n* = 66 neuro-cardiac contacts from three different rat hearts). [Colour figure can be viewed at wileyonlinelibrary.com]

distance between varicosities), in the presence *vs.* absence of NGF in the culture medium, at either time point (Fig. 3*C–E*).

Moreover, morphometric analyses demonstrated that SN varicosities grew in time (area of TH+ enlargements) to a similar extent in the absence or the presence of exogenous NGF (Fig. 3*D*). The only difference, which was evident already at qualitative level, in the two conditions was the increased neuronal axonal sprouting, in NGF-added co-cultures (Fig. 3*C*), consistent with the

effect of diffuse stimulation of neurons with NGF in the medium. These results suggest that CMs represent the cell source of neurotrophin, required to sustain neurons. In further support, quantitation of NGF in protein extracts from CMs, maintained in culture for up to 14 days, demonstrated that cell maturation is accompanied by opposite changes in the relative content of pro-NGF *vs.* mature NGF forms, the latter significantly increasing with time (Fig. 4*A*). Such results, suggesting progressive maturation of post-transcriptional processing

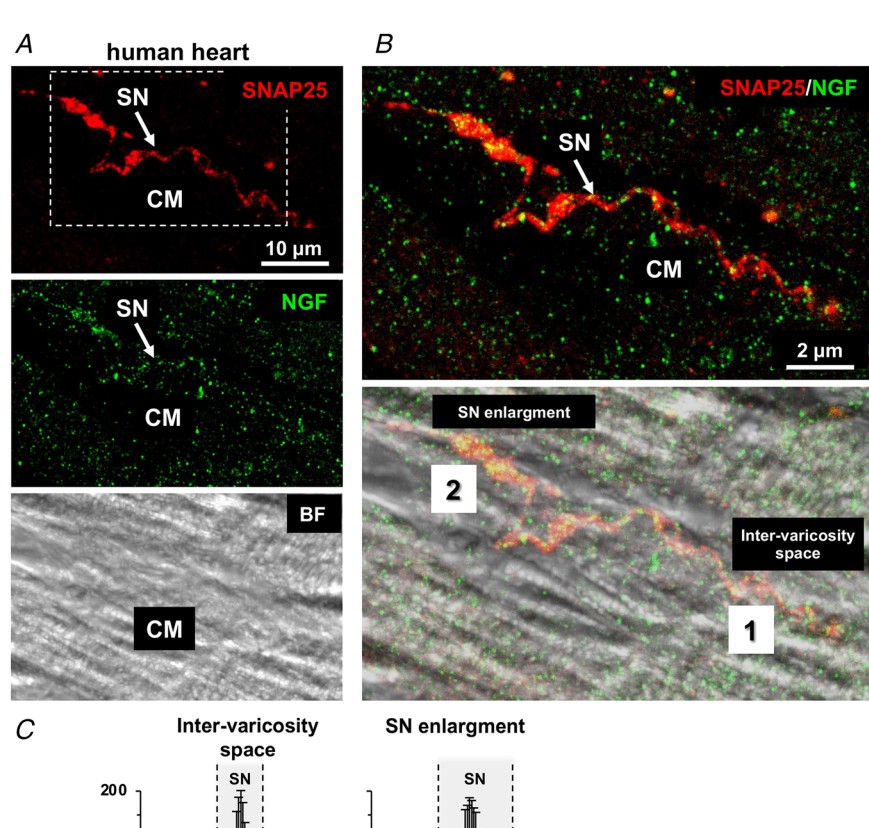

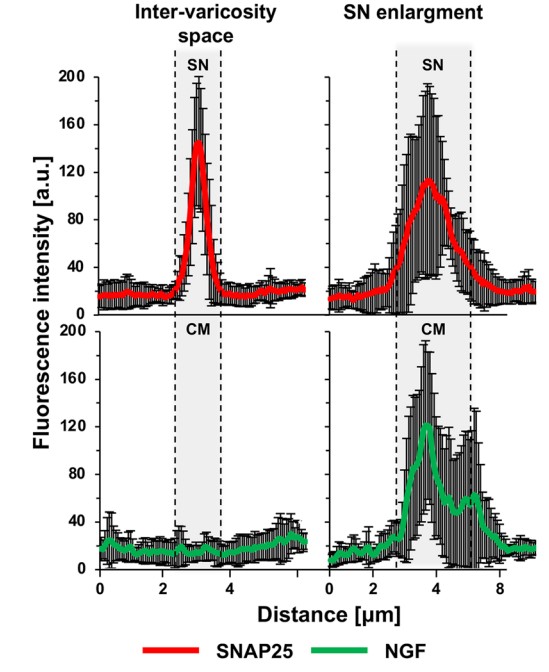

**Figure 2. Nerve growth factor mainly concentrates in the cardiomyocyte submembrane portion contacted by sympathetic neurons in human hearts**
*A* , confocal immunofluorescence (IF) analysis of human heart sections, co-stained with antibodies against SNAP25 (top panel) and nerve growth factor (NGF) (middle panel). The bottom panel shows the bright field image. *B*, magnification of the white box in (*A*), showing the merged fluorescence and bright field images. SN, sympathetic neuron; CM, cardiomyocyte. Arrows indicate NGF puncta in the neuronal process. *C*, quantification of the fluorescence intensity of SNAP25 (red signal) and NGF (green signal) in correspondence with the neuronal varicosity (2) or the inter-varicosity space (1). Bars represent SD (*n* = 15 neuro-cardiac contacts from two different human hearts). [Colour figure can be viewed at wileyonlinelibrary.com]

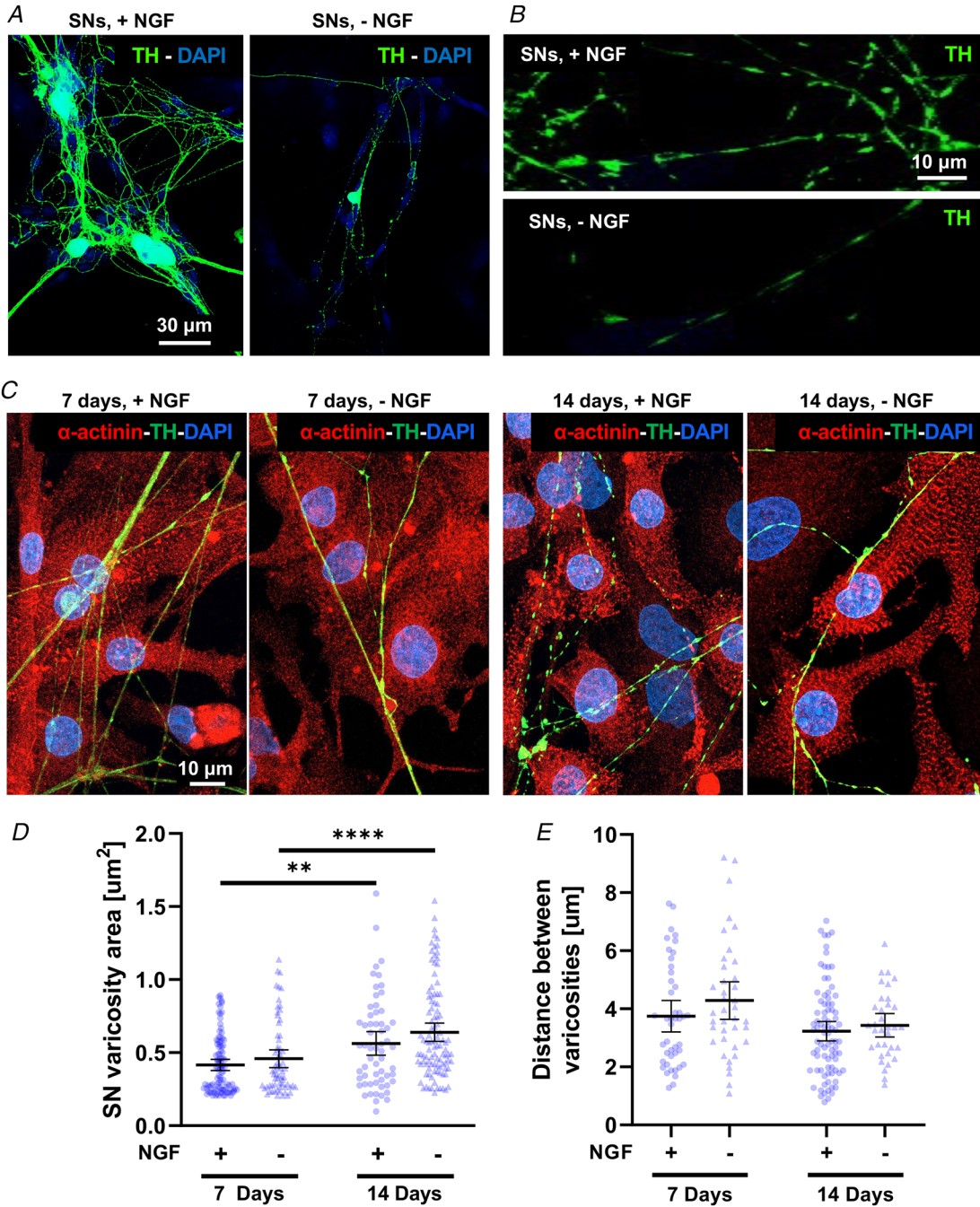

**Figure 3. Time- and neurotrophin-dependency of sympathetic neuron maturation in co-culture**
*A–B*, confocal immunofluorescence (IF) analysis of 7-day cardiac sympathetic neurons (cSNs), isolated from the superior cervical and stellate ganglia of neonatal rats, cultured in the presence (+NGF) or the absence (-NGF) of nerve growth factor (NGF). Cells were stained with an antibody against tyrosine hydroxylase (TH). Nuclei were counterstained with DAPI. *C*, confocal IF analysis of 7-day (left panels) *vs*. 14-day (right panels) SN/ cardiomyocyte (CM) co-cultures, maintained in the presence or in the absence of NGF. Cells were co-stained with antibodies against $\alpha$-actinin and TH. Nuclei were counterstained with DAPI. *D–E*, quantification of SN varicosity area (*D*) and interindividual distance between varicosities (*E*) in cSNs co-cultured with CMs, in the absence (-) *vs*. the presence (+) of NGF in the culture medium. Data distribution is represented by the individual values. Mean and error bars, representing 95% confidence intervals, are shown. Differences among groups were determined using the Mann–Whitney test. (**, $P < 0.01$; ****, $P < 0.0001$; (+) 7 days $n = 101$ and $n = 44$; (+) 14 days $n = 60$ and $n = 85$; (-) 7 days $n = 67$ and $n = 39$; (-) 14 days $n = 102$ and $n = 34$ varicosities for each group. Three independent cell preparations were analysed). [Colour figure can be viewed at wileyonlinelibrary.com]

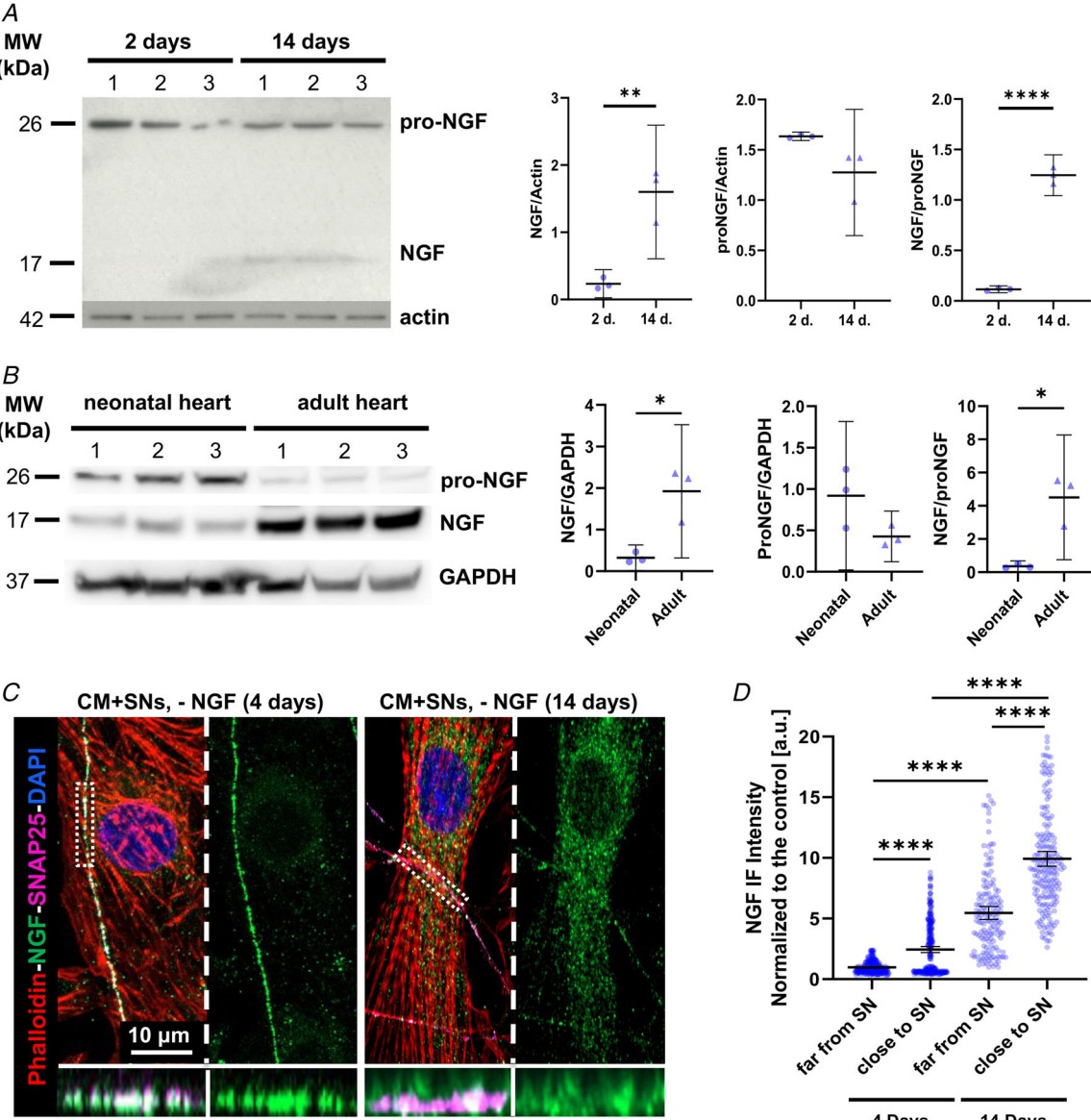

**Figure 4. Time dependency of nerve growth factor content and distribution in cardiomyocytes**

*A*, western blotting of pro-nerve growth factor (NGF) and mature NGF (left panel) on protein extracts from cardiomyocytes (CMs) maintained for 2 or 14 days in culture. Actin was used as loading control. The right panels show the relative densitometry. Data distribution is represented by the individual values. Mean and error bars, representing 95% confidence intervals, are shown. Differences among groups were determined using an unpaired *t* test or Mann–Whitney test. (\*\*, *P* < 0.01; \*\*\*\*, *P* < 0.0001. Three independent experiments were performed). *B*, western blotting of pro-NGF and mature NGF (left panel) on protein extracts from neonatal (P7) and adult (3 months old) rat hearts. GAPDH was used as loading control. The right panel shows the relative densitometry. Data distribution is represented by the individual values. Mean and error bars, representing 95% confidence intervals, are shown. Differences among groups were determined using an unpaired *t* test. (\*, *P* < 0.05; *n* = 3 hearts for each group. Three independent experiments were performed). *C*, confocal immunofluorescence analysis on 4 *vs*. 14-day sympathetic neuron (SN)/CM co-cultures maintained in the absence of exogenous NGF in the culture medium. Cells were co-stained with antibodies against NGF and SNAP25. CMs were stained with Alexa633-conjugated phalloidin, while nuclei were counterstained with DAPI. Bottom panels show resliced images of the boxed area highlighting the intercellular interface. *D*, comparison of the fluorescence intensity of NGF signal in CM portions close to, or far from, the contacting innervating neuronal process, in the culture conditions described in (*C*). Data distribution is represented by the individual values. Mean and error bars, representing 95% confidence intervals, are shown. Differences among groups were determined using the Mann–Whitney test. (\*\*\*\*, *P* < 0.0001; 4 days: *n* = 267 (far) and 296 (close); 14 days: *n* = 150 (far) and 192 (close) areas analysed/group. Three independent cell preparations were analysed). [Colour figure can be viewed at wileyonlinelibrary.com]

of the neurotrophin, support the time-dependency of SN maturation elicited by targeted CMs. In addition, they are in line with the findings in postnatal heart development, which show progressive increase in cardiac content of mature NGF between partially innervated neonatal *vs.* fully innervated adult hearts (Fig. 4*B*). Notably, confocal IF demonstrated that NGF was detectable in CMs in both early and mature co-cultures, with a vesicular distribution around the cell nucleus, consistent with the production and maturation sites of the neurotrophin, and tended to accumulate underneath the CM membrane portion contacted by the neuronal process (Fig. 4*C* and *D*).

We subsequently sought to determine whether target-derived neurotrophic effects depended solely on NGF availability or if they required cell-specific structured interactions. We thus compared neuronal morphology and NGF uptake in co-cultures set up between SNs and either CMs or CFs, the latter previously shown (Mias et al., 2013), and confirmed by us, to synthesize a high amount of mature NGF (Fig. 5*A*). Interestingly, in 14-day co-cultures, the mean NGF immunoreactivity in the portion of SN processes contacting CFs (Fig. 5*B*) was lower than that measured in SN processes innervating CMs (Fig. 5*C*). This result suggests that, despite CFs having high availability of NGF, efficiency of its uptake by SNs is reduced. In line with this, in SN/CF co-cultures, removal of NGF caused a significant decrease in neuronal density (more than $50 \pm 4\%$ in 7-day cultures; more than $80 \pm 7\%$ in 14-day cultures), accompanied by a reduction in the size of neuronal varicosities, when compared with those from SNs contacting CMs, in the same culture conditions (Fig. 5*D* and *E*).

To further isolate the macroscopic effect of prolonged cell-to-cell interaction on the maturation of intercellular contact sites, we took advantage of our previous demonstration that, contrary to CMs, CFs are unable to establish stable and structured interaction with innervating SNs (Pianca et al., 2019). We thus assessed the topography of SN varicosities, in SN/CM and SN/CF co-cultures, in the three dimensions (3D), using SICM. SICM allows the surface area and volume of the neuronal varicosity in contact with the target cell (CM or CF) to be quantitated at high resolution. Coordinates of the single voxel of the scanned surface (i.e. x,y,z) are assigned to a 3D matrix, rendered as an image with specific analysis software. The output which can be extrapolated includes the crude morphology of the scanned object (e.g. varicosity), the direct measure of the object's height, its surface area and calculated volume. In this study, we firstly focused on SN/CM co-cultures, at different time points (i.e. 2, 4, 10 and 14 days), maintained in the absence of NGF. From the morphological point of view, the initial (at 2 days) tubular shape of neuronal processes developed, from 4 days onwards, enlargements of progressively higher surface area and volume, which stand out from the CM layer (height), achieving the typical pearl-necklace morphology of SNs (Fig. 6*A* and *B*). As shown previously (Prando et al., 2018), growth of varicosities was maximal at 14 days in culture (Fig. 6*C*), while no further changes occurred with more prolonged time in culture. Remarkably, the trophic effect of CMs on contacting varicosities was independent of the presence of NGF in the culture medium, as no differences in morphology and size were observed with or without the neurotrophin in the culture medium (Fig. 6*D*).

On the contrary, the varicosity size of neuronal processes contacting CFs did not increase in time and, after the same period in culture, all morphometric parameters remained significantly lower than those calculated in neurons innervating CMs (Fig. 6*E*).

Altogether, these data suggest that CMs may be a source of neurotrophin for the innervating neurons, and strongly support the hypothesis that retrograde 'target cell-to-SN' neurotrophic signalling requires the establishment of structured and stable intercellular contact sites (i.e. NCJs).

### Cardiomyocyte-derived NGF is essential to sustain the innervating sympathetic neurons

If CM-released NGF were necessary to sustain innervating SNs, we would expect interference with its production by CMs to result in degeneration of the innervating neurons. We thus transfected CMs with plasmids encoding for either siRNA for NGF (si44), designed to ablate NGF expression, or a scramble plasmid, as control. A GFP-encoding plasmid was combined with each siRNA to identify successfully transfected CMs (Fig. 7). Our results show that si44 caused a $(61.05 \pm 5.85\%)$ decrease in NGF expression, compared with controls, and did not interfere with the expression of other CM neurotrophins, such as NT3 (Fig. 7*A*). In addition, cell transfection did not affect CM viability and morphology, since no significant differences in sarcomere organization (Fig. 7*B*), CM area or density were observed (cell area, untransfected CMs: $958 \pm 284$ *vs.* scramble CMs: $1176 \pm 462$ *vs.* NGF siRNA CMs: $1117 \pm 592$, in $\mu\text{m}^2$, data expressed as means $\pm$ SD; $P = $ ns) (cell density, untransfected CMs: $394 \pm 115$ *vs.* scramble CMs: $410 \pm 76$ *vs.* NGF siRNA: $341 \pm 95$, in cells/$\text{mm}^2$, data are expressed as means $\pm$ SD; $P = $ ns). Consistent with our hypothesis, SN innervating NGF-silenced CMs appeared fragmented and had smaller TH-marked varicosities, compared with controls (Fig. 7*C* and *D*). Notably, all these effects were prevented by NGF addition to the culture medium.

These results prove that SN trophism depends on NGF directly provided by the innervated CMs.

## Maturation of intercellular 'sympathetic neuron-cardiomyocyte' contacts parallels TrkA directional signalling

It is well-accepted that NGF plays its role in sustaining SN development, growth and survival, through activation of the high affinity neurotrophin receptor, TrkA (Chao, 2003; Reichardt, 2006; Zweifel et al., 2005). Consistently, confocal IF showed that SNs expressed, in both early (4 day) and long-term (14 day) co-cultures, the NGF-receptor TrkA (Fig. 8*A*). This evidence, together with the preferential accumulation of NGF underneath

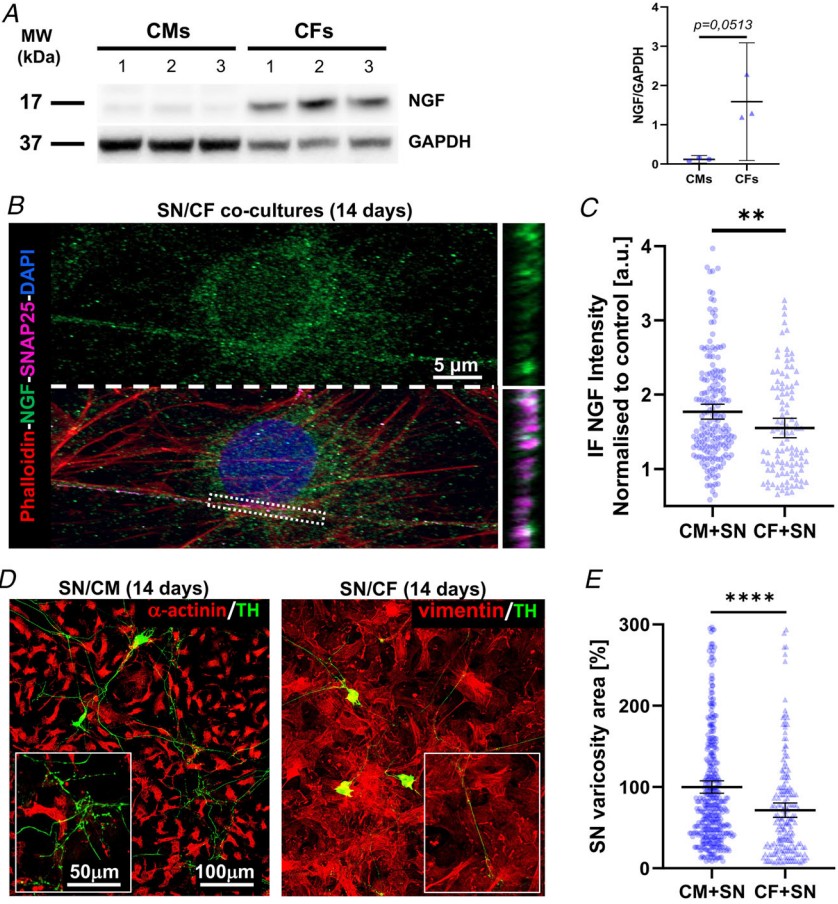

**Figure 5. Sympathetic neurons co-cultured with cardiac fibroblasts require exogenous nerve growth factor**

*A* , western blotting of nerve growth factor (NGF) protein content on extracts from cardiac fibroblasts (CFs) and cardiomyocytes (CMs), maintained in culture for 14 days, in the absence of exogenous NGF. Actin was used as loading control. The right panel shows the relative densitometry. Data distribution is represented by the individual values. Mean and error bars, representing 95% confidence intervals, are shown. Differences among groups were determined using an unpaired *t* test. (Three independent experiments were performed). *B*, confocal immuno-fluorescence (IF) analysis of 14-day sympathetic neuron (SN)/CF co-cultures maintained in the absence of exogenous NGF in the culture medium. Cells were co-stained with antibodies against SNAP25 and NGF and counterstained with Alexa633-conjugated phalloidin and DAPI. Bottom panels show resliced images of the boxed area highlighting the intercellular interface. *C*, quantification of the mean NGF fluorescence intensity in the portion of SN processes contacting CMs (black bars) *vs.* CFs (grey bars). Data distribution is represented by the individual values. Mean and error bars, representing 95% confidence intervals, are shown. Differences among groups were determined using the Mann–Whitney test. (**, $P < 0.01$; SN/CM: $n = 186$ and SN/CF: $n = 104$ areas analysed/group. Three independent cell preparations were analysed). *D*, confocal IF analysis of 14-day SN/CM (left panel) *vs.* SN/CF (right panel) co-cultures maintained in the absence of exogenous NGF. Cells were co-stained with antibodies against tyrosine hydroxylase (TH) and either anti-α-actinin (for SN/CM co-cultures) or anti-vimentin (for SN/CF co-cultures). *E*, quantification of the mean area of neuronal varicosities in contact with either CMs or CFs. To highlight the differences between the two independent populations, values of the area of SN/CF contact sites were normalized to the average area size of varicosities in contact with CM. Mean and error bars, representing 95% confidence intervals, are shown. Differences among groups were determined using the Mann–Whitney test. (****, $P < 0.0001$; $n = 24$ cell couples for each group. Three independent cell preparations were analysed). [Colour figure can be viewed at wileyonlinelibrary.com]

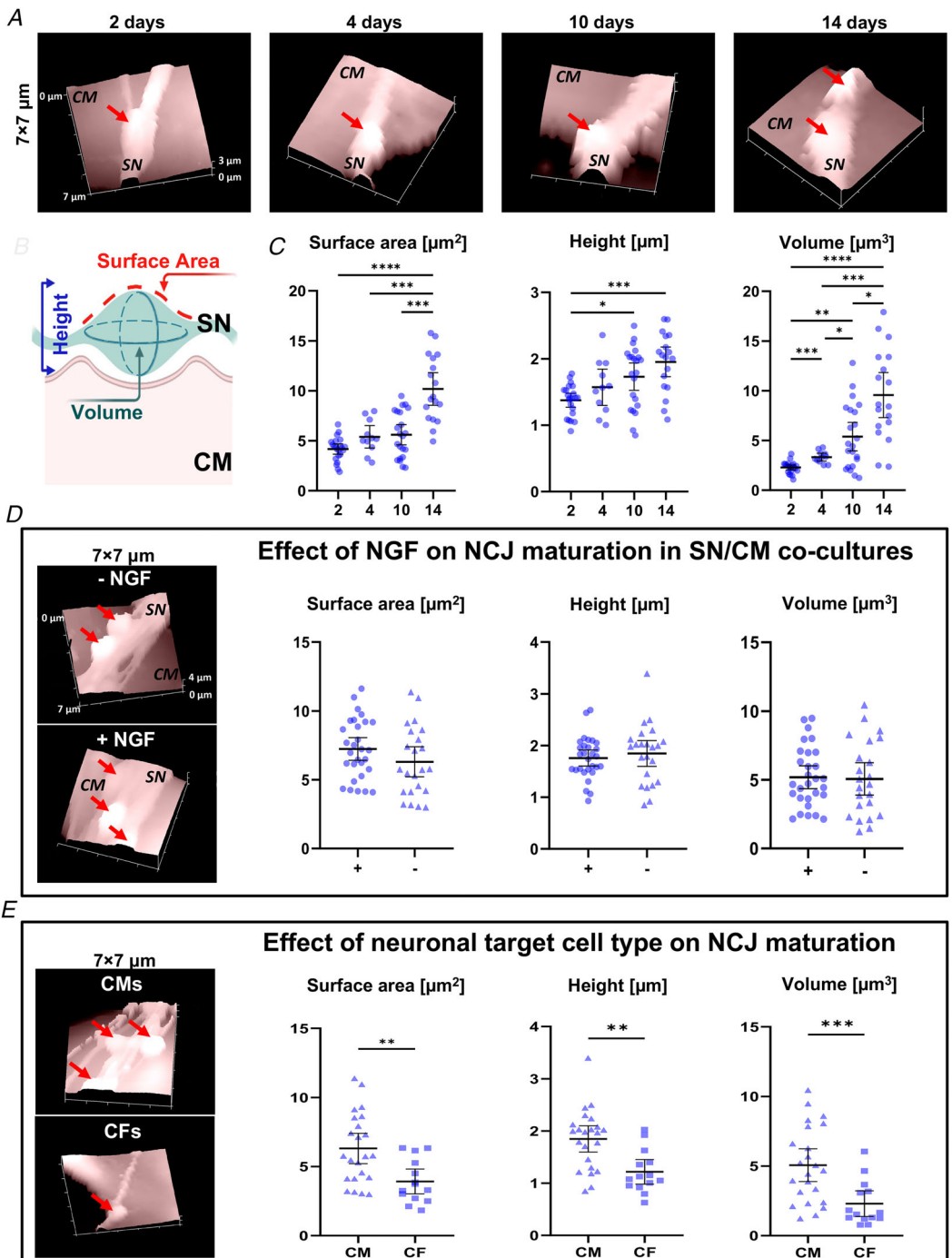

**Figure 6. Surface topography of neuronal varicosities during co-culture maturation**

*A* , representative topographical images of sympathetic neurons (SNs) in 2-, 4-, 10- and 14-day co-culture with cardiomyocyte (CM). Red arrows indicate neuronal varicosities. *B*, topographical parameters evaluated on SICM images and (*C*) relative measurements in SN/CM co-cultures analysed at different time points. Data distribution is represented by the individual values. Mean and error bars, representing 95% confidence intervals, are shown. Differences among groups were determined using the Brown–Forsythe test, with Dunnett's correction. (*, $P < 0.05$; **, $P < 0.01$; ***, $P < 0.001$; ****, $P < 0.0001$; $n$ = 15 SN/CM contacts for each group). *D*, representative SICM surface scans and quantification of relative parameters of the contact sites in 10-day SN-CM co-cultures, maintained in the absence (-) or the presence (+) of exogenous nerve growth factor (NGF). Data distribution is represented by the individual values. Mean and error bars, representing 95% confidence intervals, are shown. Differences among groups were determined using an unpaired *t* test. (*P* = ns; $n$ = 23 SN/CM contacts for each group). *E*, representative SICM surface scans and quantification of relative parameters of the contact sites in 10-day

the portion of CM membrane contacted by the neuronal process (Fig. 4*C* and *D*), suggest that the co-cultures replicate well the features observed in the intact myocardium, and are thus suited to interrogate the mechanisms of intercellular neurotrophin signalling, *in vitro*. Moreover, based on the data acquired thus far, we expect that activation of neuronal TrkA initiates at the neuro-cardiac interface and that the efficiency of intercellular signalling increases with NCJ maturation.

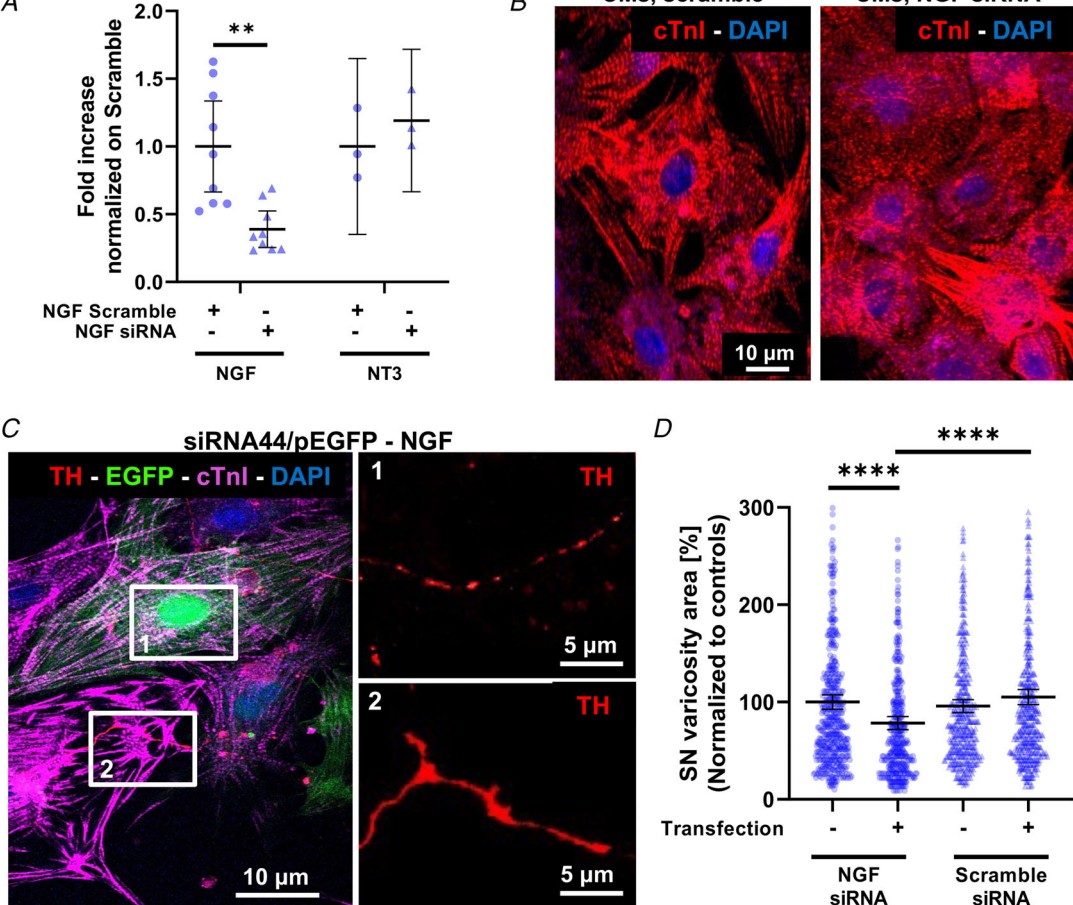

**Figure 7. Effect of nerve growth factor silencing in cardiomyocytes on co-cultured sympathetic neurons**
*A*, RTqPCR on extracts from cultured cardiomyocytes (CMs) transfected with scramble siRNA (white bars) or nerve growth factor (NGF) siRNA (black bars). Data distribution is represented by the individual values. Mean and error bars, representing 95% confidence intervals, are shown. Differences among groups were determined using an unpaired *t* test. (**, $P < 0.01$; $n = 6$ samples for each group in three independent cell preparations). *B*, confocal immunofluorescence (IF) of cultured CMs transfected with either scramble siRNA or NGF siRNA. Cells were stained with an antibody against cardiac troponin I (cTnI). Nuclei were counterstained with DAPI. *C*, confocal IF of 7-day SN/CM co-cultures in which CMs were co-transfected with green fluorescent protein GFP and NGF siRNA. Cells were co-stained with antibodies against tyrosine hydroxylase (TH) and cTnI. Nuclei were counterstained with DAPI. Images on the right are high magnifications of boxed areas on the left panel and show neuronal processes innervating, respectively, GFP+/NGF-silenced (1) or control (2) CMs. *D*, quantification of the mean area of TH+ sites contacting un-transfected, scramble-siRNA transfected or NGF-silenced CMs, in 7-day co-cultures maintained in the absence of exogenous NGF. Data distribution is represented by the individual values. Mean and error bars, representing 95% confidence intervals, are shown. Differences among groups were determined using the Mann–Whitney test. (****, $P < 0.0001$; $n > 400$ neuronal processes for each condition, in three independent cell preparations). [Colour figure can be viewed at wileyonlinelibrary.com]

To quantitate activation of NGF/TrkA signalling in co-cultures, we infected SNs with a construct encoding the fluorescent fusion protein, TrkA-DsRed, and used confocal time-lapse imaging to monitor TrkA trafficking, as an effect of receptor activation by NGF (Fig. 8*B*). We thus compared neuronal TrkA-DsRed2 movements in early *vs.* mature SN/CM co-cultures, which were quantified through kymograph analysis (Fig. 8*C*). Consistent with the literature (Zweifel et al., 2005), both stationary and bidirectional (i.e. anterograde and retrograde) moving red fluorescent vesicles were detected in SN processes. While in early co-cultures the relative fractions of anterograde and retrograde movements of TrkA dots were comparable, and higher than that of

stationary dots, we observed a significant increase in the percentage of directionally moving and stationary TrkA dots in mature co-cultures (Fig. 8*D*). These results indicate increased NGF/TrkA signalling from the distal portions of the neuron to the soma, which may depend on higher NGF availability in target cells and maturation of the NCJ allowing more efficient intercellular communication (see Figs 4 and 6). In line with the reduced neurotrophic effects of CFs, which lack structured NCJs (see above), on innervating neurons, directional TrkA trafficking was reduced by more than 70% in SN/CF co-cultures (retrograde movements, on CMs: $52.61 \pm 31.93$ ($n = 41$) *vs.* on CFs: $12.90 \pm 11.66$ ($n = 20$); $P < 0.0001$).

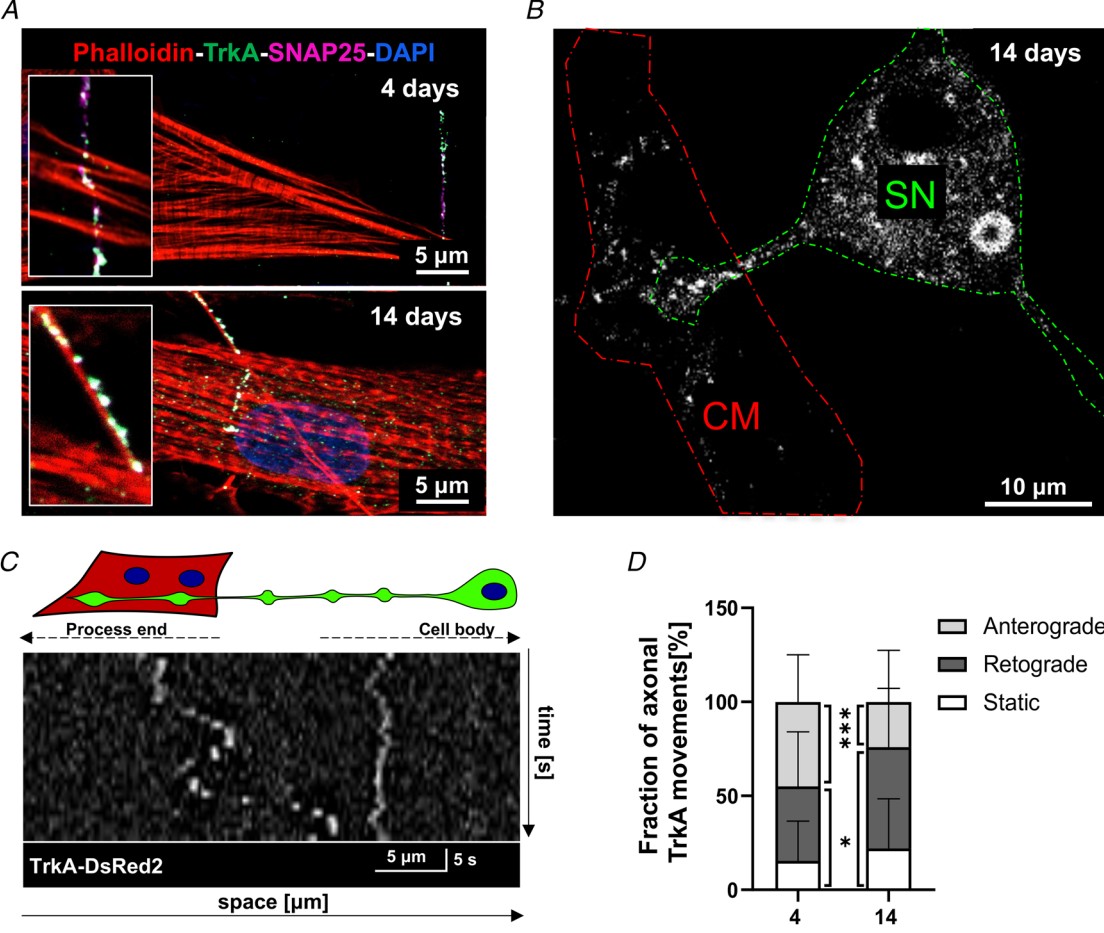

**Figure 8. TrkA signalling assay in sympathetic neurons during time in co-culture**
*A*, confocal immunofluorescence (IF) analysis of 4-day (top panel) *vs.* 14-day (bottom panel) co-cultures co-stained with antibodies against TrkA and SNAP25. Cells were co-stained with Alexa633-conjugated phalloidin. Nuclei were counterstained with DAPI. *B*, confocal IF of sympathetic neuron (SN)/cardiomyocyte (CM) co-culture infected with an adenoviral vector encoding TrkA-DsRed2. The dashed lines indicate SN (green) and innervated CM (red). *C*, representative kymograph of TrkA-DsRed2 dots in one neuronal process of a 4-day SN/CM co-culture. *D*, quantification of TrkA-DsRed2 trafficking in SN processes contacting CMs, in early (4 days) and mature (14 days) co-cultures. Values are indicated as means $\pm$ SD. Differences among groups were determined using the Mann–Whitney test. (*, $P < 0.05$; ***, $P < 0.001$; (4 days): $n = 47$ and (14 days): $n = 40$ cells per group. Three independent cell preparations were analysed). [Colour figure can be viewed at wileyonlinelibrary.com]

## The neuro-cardiac junction defines a high [NGF] extracellular signalling domain

The results collected thus far prompted us to investigate the biophysical mechanism underscoring CM-SN signalling, and we considered two hypotheses: (i) CMs sustain neuronal viability and trophism by releasing diffusely high amounts of NGF or (ii) CMs feed neurons selectively at the single contact site with SN varicosities by releasing NGF in a targeted and efficient way at the NCJ. To verify whether CMs were able to release enough NGF in the medium to sustain SN viability, we collected the medium conditioned by CMs and used it in a pure SN culture. Interestingly, CM-conditioned medium caused a ($55 \pm 3\%$) reduction in neuronal density, which was comparable to the effect of NGF deprivation on SNs alone. In agreement with this result, [NGF] in the CM-conditioned medium was about 1000-fold lower ($1.61 \pm 1$, in pg/ml) than the minimal concentration required for neuronal survival. These results rule out the possibility that the bulk NGF amount released by CMs is sufficient to sustain SNs, and suggest a model whereby elevated [NGF], activating TrkA signalling, is locally achieved at the SN/CM contact site. This hypothesis is in line with our previous demonstration that the NCJ outlines a diffusion-restricted signalling domain (Prando et al., 2018), characterized by specific protein enrichment and reduced cell-to-cell intermembrane distance (Fig. 9*A*).

To test whether intercellular NGF signalling occurred predominantly at the NCJ, we set up a NGF inhibition assay by using either: (i) an anti-NGF antibody or (ii) the membrane-permeable TrkA antagonist, k252a (Berg et al., 1992) (Fig. 9*B*). We initially tested both molecules in cultures of SNs alone, to determine which concentration caused a significant decrease of neuronal viability, in the presence of 8 n M NGF in the culture medium (Fig. 9*C*). Subsequently, we treated mature co-cultures, deprived of exogenous NGF, with the same molecules (i.e. anti-NGF, k252a) at the same concentration, and notably, while k252a substantially replicated the effect on pure SN cultures, the anti-NGF antibody had negligible effects on SN viability (Fig. 9*D*). Given the physical-chemical differences between the two compounds, and the significant steric hindrance of the anti-NGF antibody, this result suggests that the barriered CM-SN contact site opposes the permeation of the antibody in the intercellular space.

To infer the NGF concentration active in the NCJ, we compared the effect of k252a on neuronal viability in co-cultures (in which NGF was only derived from CMs), with that of k252a on pure SN cultures treated with increasing [NGF] in the culture medium. The results of such estimation suggest that the NGF concentration active at the contact site is in the order of $1.4 \pm 0.03$ n M (Fig. 9*E*).

Taken together, these results suggest that the NCJ is an isolated microenvironment protected from diffusion and characterized by high NGF concentrations.

## Discussion

This study investigates the mechanisms of neurotrophic communication between cardiac cells and heart-innervating SNs. Our results indicate that CMs provide vital support to neurons through direct exchange of NGF, which takes place locally at the neuro-cardiac interaction site. We showed that as CMs establish structured intercellular contacts, stable in time, with neurons, they guarantee more efficient neurotrophic input than CFs, throughout postnatal development. We thus refine the notion that the target organ sustains its own sympathetic innervation, by identifying the specific cell population responsible for such effect, in physiology. Our data suggest that alteration in the NCJ or in NGF signalling, resulting from primary CM injury, may thus underlie heart dysinnervation and clinically relevant cardiac dysautonomia.

In conventional neuro-cardiology, the sympathetic nervous system (SNS) is regarded as the extrinsic modulator of cardiac activity in stress conditions (Scalco et al., 2021; Zaglia & Mongillo, 2017). However, recent biotechnological advancement, which allowed the tangles of cardiac sympathetic innervation to be partially unravelled, together with a rediscovered curiosity in the physiological mechanisms of neurogenic heart regulation, led to uncovering unexpected roles of cardiac SNS, beyond ignition of the fight-or-flight reaction (for a review, see Scalco et al., 2021). An aspect which has only recently been appreciated, is that the mammalian heart is more densely innervated by SNs than expected: for example, each CM is simultaneously embraced by several neuronal processes (from three to six), which may originate from sprouted axons of the same or different neurons (Di Bona et al., 2020; Scalco et al., 2021). Furthermore, neurons are distributed with a non-random, species-specific topology, which reflects the respective cardiac electrical and mechanical properties (Pianca et al., 2019). The network architecture, established in embryonic or early postnatal development, is designed by molecular mediators released by myocardial cells, either recruiting (i.e. neurotrophic factors, such as NGF) or blocking (i.e. chemorepellent agents, such as semaphorine-3a) axonal growth in a given myocardial territory (Franzoso et al., 2016; Ieda et al., 2004; Ieda et al., 2007; Kimura et al., 2012; Lorentz et al., 2010). Additionally, we and others showed that cSN inputs continue to shape the morphological and electrophysiological properties of the adult myocardium, implying that the pattern of cardiac innervation needs to remain unchanged for physiological cardiac function (Di Bona et al., 2020; Franzoso et al., 2016;

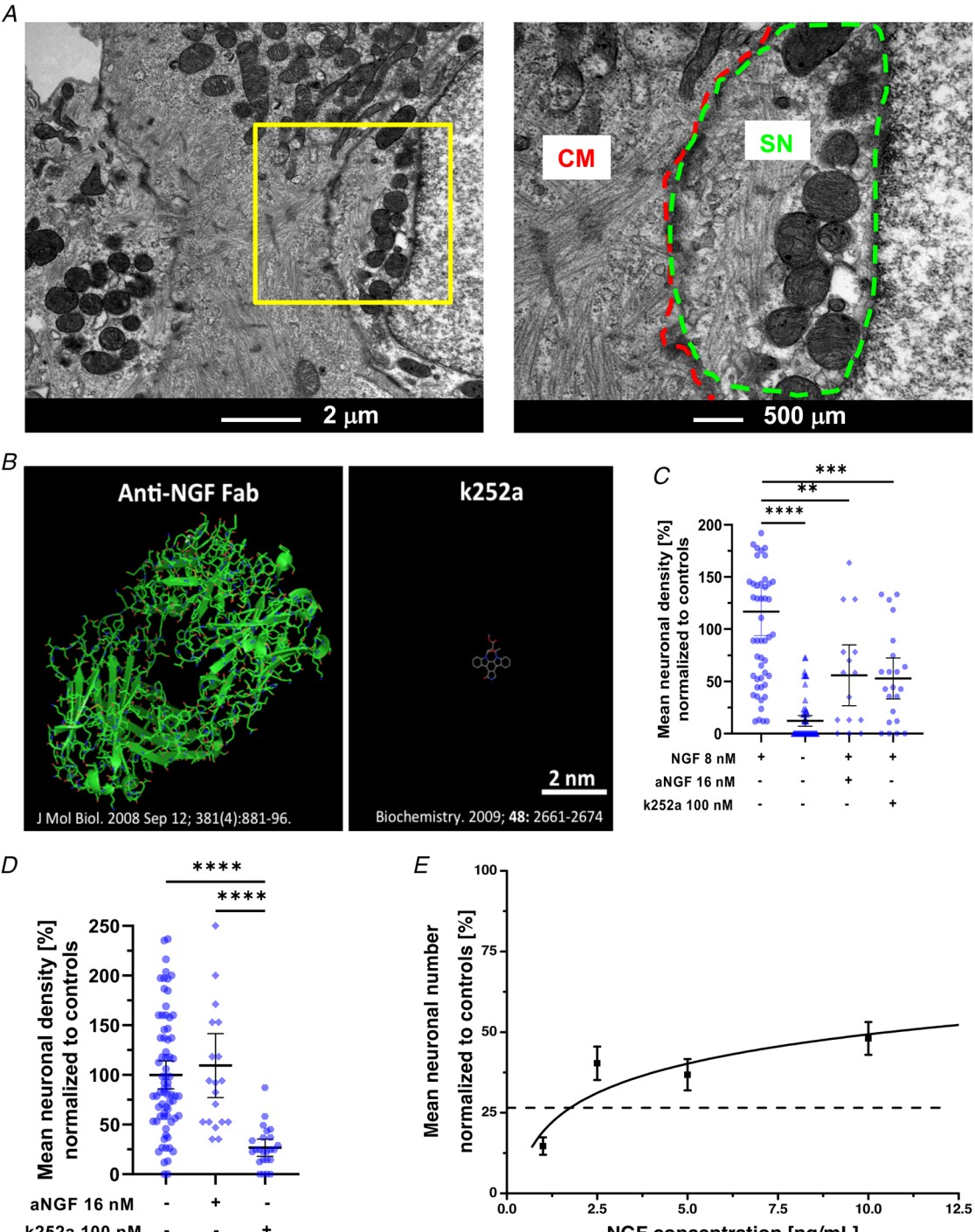

**Figure 9. Pharmacological assay of cardiomyocyte-neuron nerve growth factor signalling at the neuro-cardiac junction**

*A*, transmission electron microscopy of 14-day sympathetic neuron (SN)/cardiomyocyte (CM) co-cultures. The right panel is the magnification of the area in the left yellow inset. *B*, Jmoll 3D reconstruction of the Fab domain of an anti-nerve growth factor (NGF) IgG and of k252a. *C*, quantification of the mean neuronal density upon treatment of SN cultures with either anti-NGF or k252a. Data distribution is represented by the individual values. Mean and error bars, representing 95% confidence intervals, are shown. Differences among groups were determined using the Mann–Whitney test. (**, *P* < 0.01; ***, *P* < 0.001; ****, *P* < 0.0001; *n* = 35 fields/condition from three independent cell preparations). *D*, quantification of the mean neuronal density in SN/CM co-cultures treated as in (*C*). Data distribution is represented by the individual values. Mean and error bars, representing 95% confidence intervals, are shown. Differences among groups were determined using the Kruskal–Wallis test. (****, *P* < 0.0001;

*n* = 35 fields/condition from three independent cell preparations). *E*, dose/effect curve of NGF on neuronal density in SN cultures treated with 100 n M k252a. Interpolation was calculated using non-linear regression obtained with Microsoft Excel. Fitting line is shown (solid black). The dashed line indicates neuronal density in k252a-treated SN/CM co-cultures. Bars indicate SD. (*n* = 24 fields per conditions from three independent cell preparations). [Colour figure can be viewed at wileyonlinelibrary.com]

Habecker et al., 2016; Ieda et al., 2007; Pianca et al., 2019; Zaglia & Mongillo, 2017). Consistently, like other organs innervated by SNs, the myocardium synthesizes neurotrophins for the entire lifespan, and it is thus fundamental to understand how heart cells deliver neurotrophic signals to SNs (Franzoso et al., 2016; Habecker et al., 2008; Habecker et al., 2016). Cardiac homeostasis is thus based on finely regulated bidirectional communication between neurons and CMs, mutually necessary to ensure cell viability in one direction, and regulation in the other.

While the effects of anterograde communication between neurons and CMs has been the subject of several studies in recent decades, interest in the retrograde 'CM-to-SN' communication axis has only recently emerged. Research on this topic has been fuelled by the evidence that several diseases, primarily targeting CMs (such as MI and heart failure), lead to secondary myocardial denervation (Boogers et al., 2010; Gardner et al., 2016; Himura et al., 1993; Kimura et al., 2010; Nishisato et al., 2010; Tapa et al., 2020). Such a pathogenetic link has been attributed to failure of neurotrophic signalling from heart to neurons (Habecker et al., 2008; Habecker et al., 2016; Lorentz et al., 2010), but the underlying mechanisms, in both physiology and pathology, are still unclear. A basic unresolved question regards the dynamics of CM neurotrophin input to SNs, and the way it occurs. In other words, do CMs feed neurons organ-wide (implying NGF diffusion in the myocardial interstitium), or through direct hetero-cellular coupling (implying 'simil-synaptic' communication)?

Recently, we and others demonstrated that SNs communicate to target CMs in a synaptic fashion, by releasing noradrenaline in a diffusion-restricted intercellular domain (i.e. NCJ), allowing high [noradrenaline] to be reached at the expense of a few neurotransmitter vesicles (Prando et al., 2018; Shcherbakova et al., 2007). In the current study, we extend this discovery to the reciprocal signalling axis, demonstrating that the NCJ is also the election site of CM-dependent neuronal feeding with NGF (Fig. 10). In line with this, our confocal IF in heart slices and SN/CM co-cultures adds new pieces to the growing puzzle of molecules concentrating in correspondence with the SN/CM contact site, notably including NGF on the CM side, and its receptor TrkA, on the neuronal one. The evidence that NGF is secreted preferentially at the NCJ suggests that scaffold proteins previously shown to accumulate at the post-synaptic membrane (e.g. cadherins, $\beta$-catenin, SAP97) may also play a role in intracellular routing of the neurotrophin

(Prando et al., 2018; Shcherbakova et al., 2007). In addition, our results strongly support the hypothesis that the NCJ outlines a low volume/high [NGF] intercellular domain, allowing efficient activation of TrkA signalling, potentially at the expense of a few molecules of NGF. In further support of this, the amount of CM-derived NGF is insufficient to prevent cSN death, when added diffusely to the culture medium. Moreover, CFs despite synthesizing high amounts of NGF, fail to sustain SN viability and development in co-culture, likely due to their incapacity to establish a stable intercellular contact, and as such to direct NGF to innervating neurons. Such a central role of the NCJ in 'CM-to-SN' signalling is highlighted by the demonstration that CM-contacting SNs grew and developed indistinguishably (in terms of varicosity size and distance between varicosities) from neurons supplied with high amounts of NGF in the medium. In view of these results, we thus surmise that local communication underlies CM to SN neurotrophin signalling, leading to further mechanistic speculations on the role of the NCJ in both physiology and pathology.

Firstly, one may speculate that the amount of NGF captured by a single varicosity is not sufficient to sustain the viability of a complex post-mitotic cell, like a cSN, whose cell body is located in a ganglion distant (i.e. at the level of the neck) from the innervated myocardium (Scalco et al., 2021). However, when considering that: (i) each single varicosity locally takes up NGF from the innervated CM; and (ii) each neuronal process is made by numerous regularly distributed varicosities, innervating multiple CMs (Pianca et al., 2019; Zaglia & Mongillo, 2017), we can conjecture that the total amount of NGF reaching the neuronal soma reflects the summed contribution of innumerable varicosities. This may be thought of as a mechanism to distribute NGF supply among the different providers, protecting neurons from neurotrophin depletion subsequent to regional dysfunction of NGF-making cells. In addition, the most linear inference is that the local nature of NGF input, at specific varicosities, would be lost once TrkA reaches the nucleus. However, the evidence that while retrogradely transported NGF/TrkA exerts effects on neuronal survival and growth, stationary NGF/TrkA signalosomes operate within the single varicosity (Kuruvilla et al., 2000; Zweifel et al., 2005), prompts a mixed model whereby local neurotrophic inputs may activate both local and global cellular effects.

Secondly, although regulation of NGF transcription was not within the scope of the current research, the

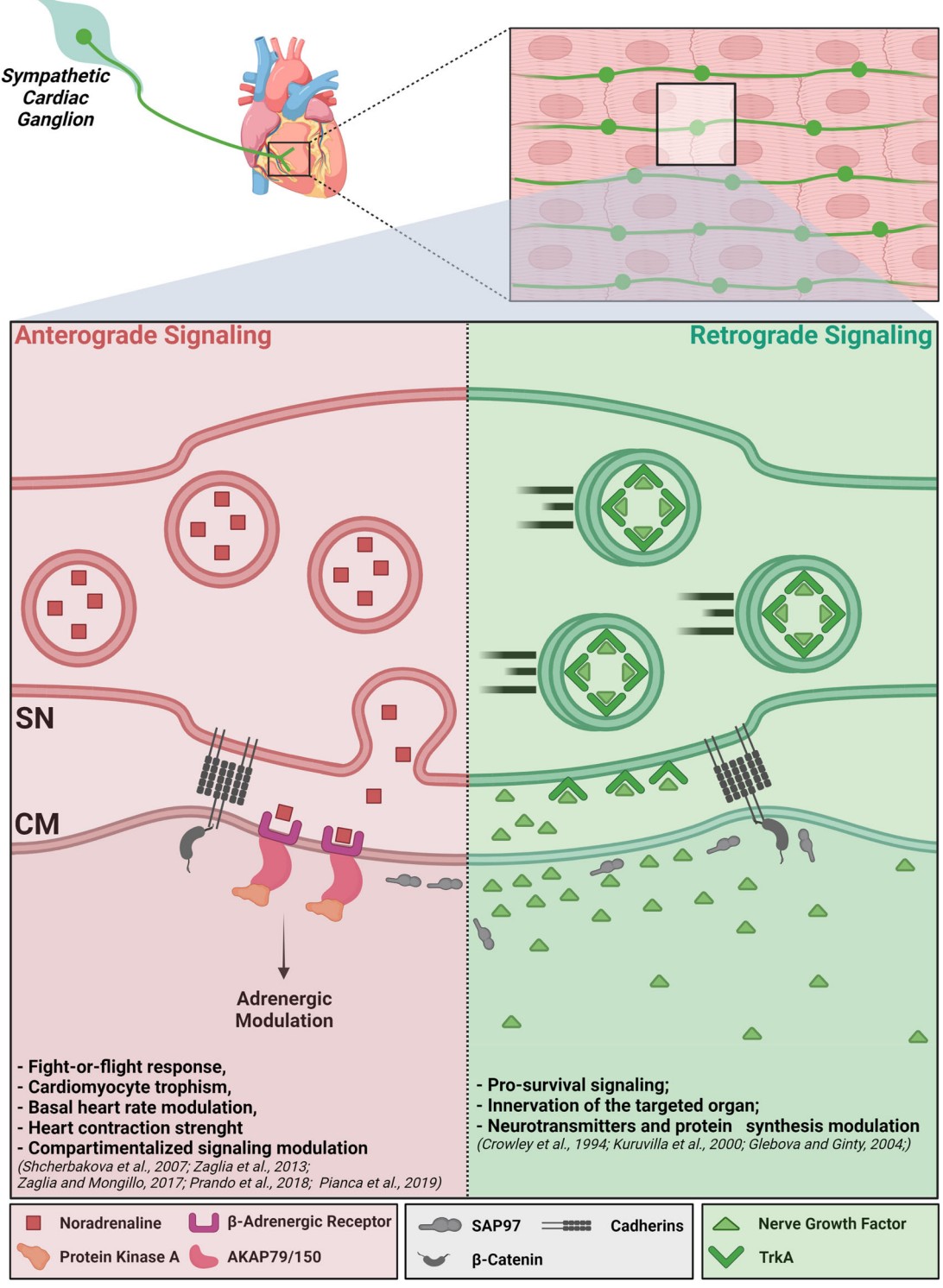

**Figure 10. The neuro-cardiac junction represents the functional unit underlying both anterograde and retrograde sympathetic neuron–cardiomyocyte communication**
*A*, roadmap of post-ganglionic sympathetic neurons (SNs), from the cervical ganglia to the myocardial interstitium.
*B*, cartoon representation of the neuro-cardiac junction is the election site of bidirectional SN–cardiomyocyte (CM) communication (created with BioRender.com). [Colour figure can be viewed at wileyonlinelibrary.com]

evidence that the NCJ is critical for both adrenergic and neurotrophin communication suggests that the two signalling axes may be cross-regulated. This speculation is based on previously published reports showing that catecholamines influence the biosynthesis of NGF, in different cell types, including CMs (Colangelo et al., 1998; Furukawa et al., 1986; Furukawa et al., 1987; Furukawa et al., 1989; Hanaoka et al., 1994; Kaechi et al., 1993). Such reciprocal interplay between CM and SNs, and the possibility that SNs might influence NGF synthesis in end organs, was proposed a few years ago (Furukawa et al., 1986), but not pursued further, to the best of our knowledge. If this model held true, it would imply a double-strand bond between CMs and innervating SNs, which is realized in the site of the NCJ.

Such bidirectional crosstalk between SN activity, CM function and neuronal viability, may have an impact on common cardiovascular therapies, and in particular the widespread use of $\beta$-AR blockers, a cornerstone drug against myocardial remodelling and arrhythmias. On the one hand, whether prolonged treatment with anti-adrenergics chronically impinged on the reciprocal CM/SN axis, it would potentially affect CM sustainment of SN viability, and thus cardiac innervation patterning. This effect could potentially reflect on heterogeneous/dysfunctional heart innervation, a condition which has, *per se*, been linked to increased arrhythmic vulnerability. Additionally, primary disruption of the NCJ, arising indirectly from other injury mechanisms (e.g. CM remodelling, myocardial ischaemia) could have devastating outcomes, as it would compromise simultaneously both neurogenic heart control and SN viability, which would worsen one another in a vicious cycle. Preservation of a healthy NCJ could therefore represent a novel therapeutic goal to be sought after in common cardiovascular disorders, including arrhythmogenic syndromes, myocardial infarction and heart failure.

To conclude, the demonstration that SN survival is strictly dependent on the directly innervated CM (Fig. 10) leads us to reformulate the concept of SNs as 'heart drivers' into that of neurons as 'CM-driven heart drivers'.

## Limitations of the work

The authors are aware that there are limitations to this study, which could be addressed by future research, and benefit from further optimization of the *in vitro* model and methods to dynamically investigate neuro-cardiac connectivity and intercellular signalling. Firstly, in the current study, we used a mixed co-culture system, in which neurons and cardiomyocyte development was not restrained by predefined patterns of cell seeding. While the use of microfluidic platforms would allow image quantitation to be simplified, especially with regards to fluorescent particle tracking, intercellular junction-independent neurotrophin and chemorepellent gradients may form in the culture, thus adding a potential bias in result interpretation. Secondly, we acknowledge that images in human heart post-mortem tissue is hardly quantifiable and comparable to murine hearts. Given the legal constraints on using human tissue, the tissue is inevitably harder to analyse, and it did in fact require a dedicated protocol. The human data, however, confirm as an important proof-of-principle, that the purported concepts are shared between rodent and human hearts.

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

## Additional information

### Data availability statement

The data that support the findings of this study are available from the corresponding author upon reasonable request.

### Competing interests

None declared.

### Author contributions

L.D. and M.F. set up the *in vitro* and *ex vivo* methodologies, performed the *ex vivo* and *in vitro* experiments, analysed the data, interpreted the results and contributed to preparation of the manuscript; A.D.B. performed the IF on human heart slices, morphometric and biochemical analyses on heart samples; N.M. performed biochemical analyses on the CF cell cultures, statistical analysis and contributed to preparation of the figures; J.L.S.A. performed the SICM experiments and analyses and discussed the data; V.P. contributed to a subset of *in vitro* experiments, PC12 cell culture and TrkA imaging; M.S. analysed the structure of the compounds used in *in vitro* studies; C.B. provided the human heart samples; G.F. critically discussed the data; H.A. shared reagents and critically discussed the data; O.M. analysed the structure of the compounds used in *in vitro* studies; J.G. supervised the SICM experiments and critically discussed the data; T.Z. and M.M. designed and supervised the study, interpreted and discussed the results and wrote the manuscript.

### Funding

This work was supported by the University of Padova (StarsWiC2017 'miniheartwork' to M.M.), STARS-SKoOP (UNIPD) to T.Z., and the British Heart Foundation (grant RG/17/13/33173) to J.G. and J.L.S.A.

### Acknowledgements

The authors are grateful to Dr Nicola Pianca for technical assistance, and Drs Raffaele Lopreiato, Giulietta Di Benedetto and Tullio Pozzan for critical discussion.

Open Access Funding provided by Universita degli Studi di Padova within the CRUI-CARE Agreement.

### Keywords

cardiac sympathetic neurons, cardiomyocytes, nerve growth factor, neuro-cardiac junction, nerve growth factor receptor

## Supporting information

Additional supporting information can be found online in the Supporting Information section at the end of the HTML view of the article. Supporting information files available:

**Statistical Summary Document**
**Peer Review History**

