## [Peer Review History · The Journal of Physiology]

Nerve Growth Factor transfer from cardiomyocytes to innervating sympathetic neurons activates TrkA receptors at the neuro-cardiac junction

Marco Mongillo, Lolita Dokshokova, Mauro Franzoso, Anna Di Bona, Nicola Moro, Jose L. Sanchez-Alonso, Valentina Prando, Michele Sandre, Cristina Basso, G Faggian, Hugues Abriel, Oriano Marin, Julia Gorelik, and Tania Zaglia
DOI: 10.1113/JP282828

Corresponding author(s): Marco Mongillo (marco.mongillo@unipd.it)

The following individual(s) involved in review of this submission have agreed to reveal their identity: Ken D O'Halloran (Referee #3)

Review Timeline:

Submission Date:	11-Jan-2022
Editorial Decision:	31-Jan-2022
Revision Received:	24-Feb-2022
Editorial Decision:	01-Mar-2022
Revision Received:	18-Mar-2022
Accepted:	28-Mar-2022

Senior Editor: Bjorn Knollmann

Reviewing Editor: Jian Shi

Transaction Report:

Dear Professor Mongillo,

Re: JP-RP-2022-282828 "Cardiomyocytes nourish innervating sympathetic neurons at the neuro-cardiac junction" by Marco Mongillo, Lolita Dokshokova, Mauro Franzoso, Anna Di Bona, Nicola Moro, Jose L. Sanchez-Alonso, Valentina Prando, Michele Sandre, Cristina Basso, G Faggian, Hugues Abriel, Oriano Marin, Julia Gorelik, and Tania Zaglia

Thank you for submitting your manuscript to The Journal of Physiology. It has been assessed by a Reviewing Editor and by 2 expert Referees and I am pleased to tell you that it is considered to be acceptable for publication following satisfactory revision.

The reports are copied at the end of this email. Please address all of the points and incorporate all requested revisions, or explain in your Response to Referees why a change has not been made.

NEW POLICY: In order to improve the transparency of its peer review process The Journal of Physiology publishes online as supporting information the peer review history of all articles accepted for publication. Readers will have access to decision letters, including all Editors' comments and referee reports, for each version of the manuscript and any author responses to peer review comments. Referees can decide whether or not they wish to be named on the peer review history document.

Authors are asked to use The Journal's premium BioRender (<https://biorender.com/>) account to create/redrawn their Abstract Figures. Information on how to access The Journal's premium BioRender account is here: <https://physoc.onlinelibrary.wiley.com/journal/14697793/biorender-access> and authors are expected to use this service. This will enable Authors to download high-resolution versions of their figures.

I hope you will find the comments helpful and have no difficulty returning your revisions within 4 weeks.

Your revised manuscript should be submitted online using the links in Author Tasks Link Not Available.

Any image files uploaded with the previous version are retained on the system. Please ensure you replace or remove all files that have been revised.

REVISION CHECKLIST:

- Article file, including any tables and figure legends, must be in an editable format (eg Word)
- Abstract figure file (see above)
- Statistical Summary Document
- Upload each figure as a separate high quality file
- Upload a full Response to Referees, including a response to any Senior and Reviewing Editor Comments;
- Upload a copy of the manuscript with the changes highlighted.

- A potential 'Cover Art' file for consideration as the Issue's cover image;
- Appropriate Supporting Information (Video, audio or data set https://jp.msubmit.net/cgi-bin/main.plex?form_type=display_requirements#supp).

To create your 'Response to Referees' copy all the reports, including any comments from the Senior and Reviewing Editors, into a Word, or similar, file and respond to each point in colour or CAPITALS and upload this when you submit your revision.

I look forward to receiving your revised submission.

If you have any queries please reply to this email and staff will be happy to assist.

Yours sincerely,

REQUIRED ITEMS:

-You must start the Methods section with a paragraph headed Ethical Approval. A detailed explanation of journal policy and regulations on animal experimentation is given in Principles and standards for reporting animal experiments in The Journal of Physiology and Experimental Physiology by David Grundy J Physiol, 593: 2547-2549. doi:10.1113/JP270818.). A checklist outlining these requirements and detailing the information that must be provided in the paper can be found at: <https://physoc.onlinelibrary.wiley.com/hub/animal-experiments>. Authors should confirm in their Methods section that their experiments were carried out according to the guidelines laid down by their institution's animal welfare committee, and conform to the principles and regulations as described in the Editorial by Grundy (2015). The Methods section must contain details of the anaesthetic regime: anaesthetic used, dose and route of administration and method of killing the experimental animals.

-The Journal of Physiology funds authors of provisionally accepted papers to use the premium BioRender site to create high resolution schematic figures. Follow this link and enter your details and the manuscript number to create and download figures. Upload these as the figure files for your revised submission. If you choose not to take up this offer we require figures to be of similar quality and resolution. If you are opting out of this service to authors, state this in the Comments section on the Detailed Information page of the submission form.

-Please upload separate high-quality figure files via the submission form.

-You must upload original, uncropped western blot/gel images (including controls) if they are not included in the manuscript. This is to confirm that no inappropriate, unethical or misleading image manipulation has occurred <https://physoc.onlinelibrary.wiley.com/hub/journal-policies#imagmanip> These should be uploaded as 'Supporting information for review process only'. Please label/highlight the original gels so that we can clearly see which sections/lanes have been used in the manuscript figures.

-Please ensure that the Article File you upload is a Word file.

-A Statistical Summary Document, summarising the statistics presented in the manuscript, is required upon revision. It must be on the Journal's template, which can be downloaded from the link in the Statistical Summary Document section here: https://jp.msubmit.net/cgi-bin/main.plex?form_type=display_requirements#statistics

-Please include an Abstract Figure. The Abstract Figure is a piece of artwork designed to give readers an immediate understanding of the research and should summarise the main conclusions. If possible, the image should be easily 'readable' from left to right or top to bottom. It should show the physiological relevance of the manuscript so readers can assess the importance and content of its findings. Abstract Figures should not merely recapitulate other figures in the manuscript. Please try to keep the diagram as simple as possible and without superfluous information that may distract from the main conclusion(s). Abstract Figures must be provided by authors no later than the revised manuscript stage and should be uploaded as a separate file during online submission labelled as File Type 'Abstract Figure'. Please ensure that you include the figure legend in the main article file. All Abstract Figures should be created using BioRender. Authors should use The Journal's premium BioRender account to export high-resolution images. Details on how to use and access the premium account are included as part of this email.

EDITOR COMMENTS

Reviewing Editor:

This manuscript investigated how the retrograde transport from cardiomyocytes to cardiac sympathetic neurons could be

used to stabilize neuronal structure and viability through a variety of imaging-based techniques in both human samples and murine tissues. The results indicated that cardiac sympathetic neurons relied on NGF derived from cardiomyocytes for the maintenance of their proper morphology and viability. As the referees have commented, the study is interesting with a potential impact on the field.

The referees have made some specific comments the authors need to take consideration of to further strengthen their findings. These comments include the clarification of experimental and data analysis details, addition of more experimental data, text changes in the manuscript to increase the readability and quality etc. Please make point-to-point response to all questions raised by referees.

On ethics: please clarify in the manuscript itself whether the human studies were given informed consent; whether the study was approved by local ethics committee; whether the method of animal euthanasia and terminal procedures were approved and acceptable.

On statistics: in line with the journal's statistics policy, all statistics data should be presented with SD (standard deviation) rather than SEM, please make the changes for all data. Also, please specify the statistical significance with precise value if applicable. Please also include a Statistical Summary Document.

Senior Editor:

I concur with the reviewing editor's assessment and recommendations.

Please revise the MS to conform to the statistics policy

REFeree COMMENTS

Referee #1:

The manuscript by Dokshokova, Franzoso et al aims to investigate the mechanisms of interaction between cardiomyocytes (CMs) and cardiac sympathetic neurons (SNs). The authors previous work established their interaction at the neuro-cardiac junction and use anterograde transport (SNs to CMs) to influence cardiac activity and conduction. The work presented focuses on how the reverse, retrograde transport (CMs to SNs) can be used to stabilize neuronal structure and viability. The authors investigated CM to SN transport primarily through a variety of imaging-based techniques, including confocal microscopy, Scanning Ion Conductance Microscopy, topographical tracing, and live cell imaging in isolated murine CMs and SNs. Human heart samples were used to verify the SNs to CM intercellular contacts at the neuro-cardiac junction. The results showed CM-derived Neuronal Growth Factor (NGF) was closely localized and elevated around areas innervated by SNs. Additionally, SNs showed a higher density of the NGF receptor TrkA. The authors directly tested the cell-cell interaction by probing at the dependence SNs had on CM-derived NGF by altering NGF levels the SN culture medium, as well as through genetic knockdown using NGF siRNA. The results showed SNs relied on CM-derived NGF to maintain proper morphology and viability.

Overall, this is an interesting study that highlights the symbiosis between cardiac nerves and cardiomyocytes and alludes to how cardiac injury may disrupt this communication axis and cause autonomic dysregulation. However, a few points need to be addressed that will be useful:

Major Comments:

- Can the authors clarify how they defined the inter-varicosity space and SN enlargement in a consistent and replicable manner?
- A graph comparing human and mouse data for the SNAP25 and NGF colocalization would be useful, as the colocalization appears weaker in human representative images.
- SN density (number of cell soma/area) and axonal sprouting is referred to in text but not shown.
- Fig 3A: It is unclear what timepoint these cell images were taken at, and additional quantification would be useful if possible.

- Fig 3D & Fig 5E: It is unclear how the area is normalized, is it area per nerve? Or total area of all cultured SNs?
- Fig. 8: The text mentions "TrkA trafficking was reduced by more than 70% in mature SN/CF co-cultures" without data.

Referee #2:

The manuscript investigates the relationship between sympathetic neurons and cardiac myocytes, where the authors have evidence that retrograde signalling from the myocytes via nerve growth factor (NGF) plays an important role in the fidelity of neurons.

Specifically, the authors provide anatomical evidence for the close contact of sympathetic neurons and cardiac myocytes. They also report that immunofluorescence on murine and human heart slices shows that NGF and its receptor, TrkA, accumulate respectively in the pre- and post-junctional sides of the neuro cardiac junction. By manipulating NGF they provide further evidence to suggest a retrograde signal to the neuron from the myocyte is important in maintaining the physiological integrity of the neuron itself.

The idea that NGF from myocytes impacts on neuronal viability has been observed for many years. However, this manuscript provides convincing data that retrograde paracrine signalling may be important in the process of sympathetic denervation that is a feature several common cardiovascular diseases. To this extent, this is a novel paper where the authors have robustly tested their hypothesis with modern experimental tools and provided a contextual framework with the use of human tissue to underpin their murine studies.

Specific Comments for attentions:

Title: I found the word 'nourish' a bit vague for a physiological concept. NGF and TrkA ought to feature in title since this is what the authors measured.

Abstract: what does 'CF' mean? Too many abbreviations in the abstract. Please minimize.

Introduction: A balanced introduction that outlines the motivation for the study. Would reference Herring et al 2019 Nat Rev Cardiol., which gives a more up to date review of the field regarding cardiac neurobiology and disease. p 5 line 13.

Results: Nicely presented and well documented. Please add more detail re human pm tissue. Patient detail: eg age, sex, reason for death etc.

Discussion: Well written. Consider a bit more discussion and interplay between SNS driving cardiac events and retrograde signalling via NGF. Important to place current results into context since current therapy is still aimed at blocking beta receptors and removing stellates in humans to prevent significant arrhythmia and sudden death. Please comment.

Please add a limitation section at the end of the current discussion since a lot of the data are highly variable and underpinned by much use of non-parametric statistics to make the point of significance.

The last sentence 'Reason (neurons) and sentiment (the heart) inevitably go hand in hand' is not needed. That is more a comment for an editorial.

END OF COMMENTS

Confidential Review

11-Jan-2022

'POINT-BY-POINT' RESPONSE

"Cardiomyocytes nourish innervating sympathetic neurons at the neuro-cardiac junction" JP-RP-2022-282828

EDITOR COMMENTS

Reviewing Editor:

This manuscript investigated how the retrograde transport from cardiomyocytes to cardiac sympathetic neurons could be used to stabilize neuronal structure and viability through a variety of imaging-based techniques in both human samples and murine tissues. The results indicated that cardiac sympathetic neurons relied on NGF derived from cardiomyocytes for the maintenance of their proper morphology and viability. As the referees have commented, the study is interesting with a potential impact on the field.

The referees have made some specific comments the authors need to take consideration of to further strengthen their findings. These comments include the clarification of experimental and data analysis details, addition of more experimental data, text changes in the manuscript to increase the readability and quality etc. Please make point-to-point response to all questions raised by referees.

On ethics: please clarify in the manuscript itself whether the human studies were given informed consent; whether the study was approved by local ethics committee; whether the method of animal euthanasia and terminal procedures were approved and acceptable.

On statistics: in line with the journal's statistics policy, all statistics data should be presented with SD (standard deviation) rather than SEM, please make the changes for all data. Also, please specify the statistical significance with precise value if applicable. Please also include a Statistical Summary Document.

Senior Editor:

I concur with the reviewing editor's assessment and recommendations. Please revise the MS to conform to the statistics policy.

Au: We thank the Editors and reviewers for the thorough revision of our manuscript, and for expressing interest in our work, and that the results have potential impact on the field.

We have understood the specific comments raised by the reviewers and thank for the suggestions which help readability of the work. All concerns were taken into consideration and addressed as detailed in the specific point-by-point rebuttal below.

We want to notify that during the time of the revision we repeated the WB comparing NGF content in cultured cardiomyocytes vs. cardiac fibroblasts, to improve the quality of the image in Figure 5A, which has thus been replaced with a new one. In detail, we processed the PVDF previously incubated with anti-GAPDH, which was stripped and incubated with a new batch of anti-NGF. While the quality of image has been substantially improved the message does not change.

In addition, the final cartoon (Figure 10) has been further improved to increase readability.

A paragraph of Ethical Approval and details on the strategy used for mouse sacrifice have been added in the Method section.

REFEREE COMMENTS

Referee #1:

The manuscript by Dokshokova, Franzoso et al aims to investigate the mechanisms of interaction between cardiomyocytes (CMs) and cardiac sympathetic neurons (SNs). The authors previous work established their interaction at the neuro-cardiac junction and use anterograde transport (SNs to CMs) to influence cardiac activity and conduction. The work presented focuses on how the reverse, retrograde transport (CMs to SNs) can be used to stabilize neuronal structure and viability. The authors investigated CM to SN transport primarily through a variety of imaging-based techniques, including confocal microscopy, Scanning Ion Conductance Microscopy, topographical tracing, and live cell imaging in isolated murine CMs and SNs. Human heart samples were used to verify the SNs to CM intercellular contacts at the neuro-cardiac junction. The results showed CM-derived Neuronal Growth Factor (NGF) was closely localized and elevated around areas innervated by SNs. Additionally, SNs showed a higher density of the NGF receptor TrkA. The authors directly tested the cell-cell interaction by probing at the dependence SNs had on CM-derived NGF by altering NGF levels the SN culture medium, as well as through genetic knockdown using NGF siRNA. The results showed SNs relied on CM-derived NGF to maintain proper morphology and viability.

Overall, this is an interesting study that highlights the symbiosis between cardiac nerves and cardiomyocytes and alludes to how cardiac injury may disrupt this communication axis and cause autonomic dysregulation. However, a few points need to be addressed that will be useful:

Major Comments:

i) Can the authors clarify how they defined the inter-varicosity space and SN enlargement in a consistent and replicable manner?

Au: Following the reviewer's suggestion, the paragraph entitled '**Evaluation of SN varicosity morphometry**' has been expanded, and we better explained the analytic flowchart and the parameters used to define neuronal structural entities in the images.

"Evaluation of SN varicosity morphometry. To analyse the size of varicosities, ROIs were manually drawn on TH-positive enlargements along neuronal processes and quantitated on the maximal projection image obtained from a 10-image series along the Z-axis and rendered using ImageJ. Enlargements were defined as axonal segments larger than twice as much the average axonal thickness in the same sample. Inter-varicosity distance was then measured with *Image J*, by calculating the distance along a line manually drawn between subsequent varicosities."

ii) A graph comparing human and mouse data for the SNAP25 and NGF colocalization would be useful, as the colocalization appears weaker in human representative images.

Au: We are aware that the immunofluorescence images in human myocardial sections are less detailed than those in mouse hearts, and that comparison between human and mouse data would allow additional inferences on the relative apposition of nerve and myocyte structures.

We thus understand the reviewer's concern. The difference in SNAP25 and NGF fluorescence intensity in murine vs human heart samples may be attributed to several factors:

i) firstly, while mouse samples undergone IF were cryosections from PFA-fixed hearts, human slices were obtained from formalin-fixed paraffine embedded (FFPE) heart blocks. As such, IF protocols applied in murine vs human heart samples were completely different, as described in the Method section. In detail, for the analysis of FFPE samples, we used a protocol specifically developed, as described in (*Zaglia et al. 2016*). Notably, to unmask FFPE tissues, 3 μ m thick slices were used, hampering the comparison with 10 μ m mouse cryosections;

ii) human heart blocks, analyzed in this study, were collected post-mortem, during autopsy, performed, by law, at least 48 hours after death. Thus, the quality of tissue samples is inevitably lower than that of heart biopsies.

For all these reasons a graph comparing human and mouse data would not be reliable. Human data has been added as proof of principle.

iii) SN density (number of cell soma/area) and axonal sprouting is referred to in text but not shown.

Au: We acknowledge the reviewer's suggestion, and to increase accuracy, we modified the text accordingly, by adding values and figure references, respectively.

"Based on our previous results (Prando *et al.*, 2018), we compared (early) 7 vs. (mature) 14 day co-cultures and, notably, we did not observe differences in SN density (number of cell soma/area (mm²), 7 days, +NGF:17.05±7.60 vs. -NGF:14.99 ±11.61; 14 days, +NGF:10.42±7.80 vs. 14 days, -NGF:9.39±3.29, data expressed as mean±SD; p=ns) and in the morphology of neuronal processes (i.e. size of varicosities and interindividual distance between varicosities), in the presence vs. absence of NGF in the culture medium, at either time point (**Fig.3C-E**)."

"The only difference, which was evident already at qualitative level, in the two conditions was the increased neuronal axonal sprouting, in NGF-added co-cultures (**Fig.3C**)."

iv) Fig 3A: It is unclear what timepoint these cell images were taken at, and additional quantification would be useful if possible.

Au: Amended accordingly.

"In fact, absence of NGF in the culture medium only allowed a negligible and unquantifiable amount of neurons to be detected in culture after 7 days."

v) Fig 3D & Fig 5E: It is unclear how the area is normalized, is it area per nerve? Or total area of all cultured SNs?

Au: The graph in **Figure 3D** shows the mean area of each single neuronal varicosity, which was calculated in SN-CM co-cultures at different time points, in the presence or absence of NGF.

On the contrary, **Figure 5** compares the size of varicosities in SN processes contacting CMs vs cardiac fibroblasts (CF). We apologize that the 'y axis' legend was mistakenly indicating 'normalization to CMs', while it referred to the relative comparison of the size of varicosities in contact with CFs with that of CMs. Graph caption has been modified accordingly, and details are explained in the relative Figure legend.

"**(E)** Quantification of the mean area of neuronal varicosities in contact with either CMs *or*. CFs. To highlight the differences between the two independent populations, values of the area of SN/CF contact sites were normalized to the average area size of varicosities in contact with CM. Mean and error bars, representing 95% confidence interval, are shown. Differences among groups were determined using Mann-Whitney test. (****, p<0.0001; n= 24 cell couples for each group. Three independent cell preparations were analyzed)."

vi) Fig. 8: The text mentions "TrkA trafficking was reduced by more than 70% in mature SN/CF co -cultures" without data.

Au: Data has been added in the text.

Referee #2:

The manuscript investigates the relationship between sympathetic neurons and cardiac myocytes, where the authors have evidence that retrograde signalling from the myocytes via nerve growth factor (NGF) plays an important role in the fidelity of neurons.

Specifically, the authors provide anatomical evidence for the close contact of sympathetic neurons and cardiac myocytes. They also report that immunofluorescence on murine and human heart slices shows that NGF and its receptor, TrkA, accumulate respectively in the pre- and post-junctional sides of the neuro cardiac junction. By manipulating NGF they provide further evidence to suggest a retrograde signal to the neuron from the myocyte is important in maintaining the physiological integrity of the neuron itself.

The idea that NGF from myocytes impacts on neuronal viability has been observed for many years.

However, this manuscript provides convincing data that retrograde paracrine signalling may be important in the process of sympathetic denervation that is a feature several common cardiovascular diseases. To this extent, this is a novel paper where the authors have robustly tested their hypothesis with modern experimental tools and provided a contextual framework with the use of human tissue to underpin their murine studies.

Specific Comments for attentions:

i) Title: I found the word 'nourish' a bit vague for a physiological concept. NGF and TrkA ought to feature in title since this is what the authors measured.

Au: Following the reviewer's suggestion, the title was changed in: "Nerve growth factor transfer from cardiomyocytes to innervating sympathetic neurons activates TrkA receptors at the neurocardiac junction".

ii) Abstract: what does 'CF' mean? Too many abbreviations in the abstract. Please minimize.

Au: CF means Cardiac Fibroblasts. We understand the reviewer's concern and minimized the abbreviations used in the abstract.

iii) Introduction: A balanced introduction that outlines the motivation for the study. Would reference Herring et al 2019 Nat Rev Cardiol., which gives a more up to date review of the field regarding cardiac neurobiology and disease. p 5 line 13.

Au: We are happy that the reviewer finds the introduction balanced. The reviewer is right in noticing the lack of this reference, which has been added to the text (see pag.5).

iv) Results: Nicely presented and well documented. Please add more detail re human pm tissue. Patient detail: eg age, sex, reason for death etc.

Au: Following the reviewer's suggestion, we implemented the relative Method section with patient details.

"We analyzed heart samples from three male subjects (age: 45±8 yrs) died for extra-cardiac causes (accidents),...".

v) Discussion: Well written. Consider a bit more discussion and interplay between SNS driving cardiac events and retrograde signalling via NGF. Important to place current results into context since current therapy is still aimed at blocking beta receptors and removing stellates in humans to prevent significant arrhythmia and sudden death. Please comment.

Au: Amended accordingly (see pages 20-21).

"If this model held true, it would imply a double-strand bond between CMs and innervating SNs, which realizes in the premises of the NCJ.

Such bidirectional crosstalk between SN activity, CM function and neuronal viability, may impact on common cardiovascular therapies, and in particular the widespread use of β -AR blockers, a cornerstone drug against myocardial remodeling and arrhythmias. On the one hand, whether prolonged treatment with anti-adrenergics chronically impinged on the reciprocal CM/SN axis, it would potentially affect CM sustainment of SN viability, and thus cardiac innervation patterning. This effect could potentially reflect on heterogenous/dysfunctional heart innervation, a condition which has, *per se*, been linked to increased arrhythmic vulnerability. Additionally, primary disruption of the NCJ, arising indirectly from other injury mechanisms (e.g. CM remodeling, myocardial ischemia) could have devastating outcomes, as it would compromise simultaneously both neurogenic heart control and SN viability, which would worsen one another in a vicious cycle. Preservation of a healthy NCJ could therefore represent a novel therapeutic goal to be sought after in common cardiovascular disorders, including arrhythmogenic syndromes, myocardial infarction and HF."

vi) Please add a limitation section at the end of the current discussion since a lot of the data are highly variable and underpinned by much use of non-parametric statistics to make the point of significance.

Au: Amended accordingly (see pag 21).

“LIMITATIONS OF THE WORK. The authors are aware that there are limitations in this study, which could be addressed in future research, and benefit from further optimization of the *in vitro* model and methods to dynamically investigate neuro-cardiac connectivity and intercellular signaling. Firstly, In the current study, we used a mixed co-culture system, in which neurons and cardiomyocyte development was not restrained by pre-defined patterns of cell seeding. While the use of microfluidic platforms would allow to simplify image quantitation, especially with regards to fluorescent particle tracking, intercellular junction-independent neurotrophin and chemorepellent gradients may form in the culture, thus adding a potentially bias in result interpretation. Secondly, we acknowledge that images in human heart post-mortem tissue is hardly quantifiable and comparable to murine hearts. Given the legal constraints in using human tissue, the tissue is inevitably harder to analyze, and it did in fact require a dedicated protocol. The human data, however, confirm as important proof-of-principle, that the purported concepts are shared among rodent and human hearts.”

vii) The last sentence 'Reason (neurons) and sentiment (the heart) inevitably go hand in hand' is not needed. That is more a comment for an editorial.

Au: Removed accordingly.

Dear Professor Mongillo,

Re: JP-RP-2022-282828R1 "Nerve Growth Factor transfer from cardiomyocytes to innervating sympathetic neurons activates TrkA receptors at the neuro-cardiac junction" by Marco Mongillo, Lolita Dokshokova, Mauro Franzoso, Anna Di Bona, Nicola Moro, Jose L. Sanchez-Alonso, Valentina Prando, Michele Sandre, Cristina Basso, G Faggian, Hugues Abriel, Oriano Marin, Julia Gorelik, and Tania Zaglia

Thank you for submitting your manuscript to The Journal of Physiology. It has been assessed by a Reviewing Editor and by 3 expert Referees and I am pleased to tell you that it is considered to be acceptable for publication following satisfactory revision.

The reports are copied at the end of this email. Please address all of the points and incorporate all requested revisions, or explain in your Response to Referees why a change has not been made.

NEW POLICY: In order to improve the transparency of its peer review process The Journal of Physiology publishes online as supporting information the peer review history of all articles accepted for publication. Readers will have access to decision letters, including all Editors' comments and referee reports, for each version of the manuscript and any author responses to peer review comments. Referees can decide whether or not they wish to be named on the peer review history document.

Authors are asked to use The Journal's premium BioRender (<https://biorender.com/>) account to create/redrawn their Abstract Figures. Information on how to access The Journal's premium BioRender account is here: <https://physoc.onlinelibrary.wiley.com/journal/14697793/biorender-access> and authors are expected to use this service. This will enable Authors to download high-resolution versions of their figures.

I hope you will find the comments helpful and have no difficulty returning your revisions within 4 weeks.

Your revised manuscript should be submitted online using the links in Author Tasks Link Not Available.

Any image files uploaded with the previous version are retained on the system. Please ensure you replace or remove all files that have been revised.

REVISION CHECKLIST:

- Article file, including any tables and figure legends, must be in an editable format (eg Word)
- Abstract figure file (see above)
- Statistical Summary Document
- Upload each figure as a separate high quality file
- Upload a full Response to Referees, including a response to any Senior and Reviewing Editor Comments;
- Upload a copy of the manuscript with the changes highlighted.

- A potential 'Cover Art' file for consideration as the Issue's cover image;
- Appropriate Supporting Information (Video, audio or data set https://jp.msubmit.net/cgi-bin/main.plex?form_type=display_requirements#supp).

To create your 'Response to Referees' copy all the reports, including any comments from the Senior and Reviewing Editors, into a Word, or similar, file and respond to each point in colour or CAPITALS and upload this when you submit your revision.

I look forward to receiving your revised submission.

If you have any queries please reply to this email and staff will be happy to assist.

Yours sincerely,

REQUIRED ITEMS:

-You must start the Methods section with a paragraph headed Ethical Approval. A detailed explanation of journal policy and regulations on animal experimentation is given in Principles and standards for reporting animal experiments in The Journal of Physiology and Experimental Physiology by David Grundy J Physiol, 593: 2547-2549. doi:10.1113/JP270818.). A checklist outlining these requirements and detailing the information that must be provided in the paper can be found at: <https://physoc.onlinelibrary.wiley.com/hub/animal-experiments>. Authors should confirm in their Methods section that their experiments were carried out according to the guidelines laid down by their institution's animal welfare committee, and conform to the principles and regulations as described in the Editorial by Grundy (2015). The Methods section must contain details of the anaesthetic regime: anaesthetic used, dose and route of administration and method of killing the experimental animals.

If experiments were conducted on humans confirmation that informed consent was obtained, preferably in writing, that the studies conformed to the standards set by the latest revision of the Declaration of Helsinki, and that the procedures were approved by a properly constituted ethics committee, which should be named, must be included in the article file. If the research study was registered (clause 35 of the Declaration of Helsinki) the registration database should be indicated, otherwise the lack of registration should be noted as an exception (e.g. The study conformed to the standards set by the Declaration of Helsinki, except for registration in a database.). For further information see: <https://physoc.onlinelibrary.wiley.com/hub/human-experiments>

EDITOR COMMENTS

Reviewing Editor:

Thank you for the revision. It has improved the manuscript. As the reviewers have agreed, this manuscript is interesting and would be quite influential if the following concerns can be satisfactorily addressed: ethic issues (the consent for use of human samples; sedation of adult rats prior to cervical dislocation); the potential discrepancy on NGF origin; specificity of NGF antibodies used in the current study.

Some important information pertaining to ethics and welfare is required.

Human heart samples were obtained from an archived biobank. The authors cite a national committee of Bioethics and reference the year 2006. Did the committee provide approval for the use of the samples in this study, or does the citation reference a blanket approval for such work? There is no statement on the issue of consent. If these are historical samples, or given that they are clinical samples obtained in the course of routine practice (autopsy), it may be that the committee has waived this requirement. Please provide clarity on this matter.

Please provide the approval code/number for the study in rats.

Top of page 8: Please change "sacrificed" to killed or euthanised.

Rats were killed by cervical dislocation. For rodents greater than 150g body mass, it is a requirement that animals are first sedated. Were adult rats sedated prior to cervical dislocation? Please provide details.

Senior Editor:

I concur with the reviewing editor.

REFEREE COMMENTS

Referee #1:

In this revised manuscript by Dokshokova, Franzoso et al, the authors have addressed my questions. I have no additional comments.

Referee #2:

This is much improved. Two further comments for clarification.

1. A further comment in the limitations is also required given the NGF data in neurons. In Figure 4C NGF expression appears in the neuron axon outside of the cardiomyocyte, which would imply that NGF is expressed in the varicosity, and not in the cardiomyocyte. Other studies have suggested that NGF is released by the vasculature, and that NGF is not expressed in the cardiomyocyte. How do the authors explain this discrepancy? If the authors believe that NGF is in the neurons then please provide a western blot or acknowledge the limitation here.

2. Why has the western blot band for NGF in the adult heart been cut, and what was the band above NGF, which has been cropped off in adult heart sample 1. Does this mean the NGF antibody also bound to other, non-specific targets. Please clarify?

Referee #3 (ethics review):

Thank you for submitting your manuscript to The Journal of Physiology. Some important information pertaining to ethics and welfare is required.

Human heart samples were obtained from an archived biobank. The authors cite a national committee of Bioethics and reference the year 2006. Did the committee provide approval for the use of the samples in this study, or does the citation reference a blanket approval for such work? There is no statement on the issue of consent. If these are historical samples, or given that they are clinical samples obtained in the course of routine practice (autopsy), it may be that the committee has waived this requirement. Please provide clarity on this matter.

Please provide the approval code/number for the study in rats.

Top of page 8: Please change "sacrificed" to killed or euthanised.

Rats were killed by cervical dislocation. For rodents greater than 150g body mass, it is a requirement that animals are first sedated. Were adult rats sedated prior to cervical dislocation? Please provide details.

END OF COMMENTS

1st Confidential Review

24-Feb-2022

REBUTTAL LETTER

“Nerve Growth Factor transfer from cardiomyocytes to innervating sympathetic neurons activates TrkA receptors at the neuro-cardiac junction” by *Dokshokova et al.*

Reviewing Editor Comments:

Thank you for the revision. It has improved the manuscript. As the reviewers have agreed, this manuscript is interesting and would be quite influential if the following concerns can be satisfactorily addressed: ethic issues (the consent for use of human samples; sedation of adult rats prior to cervical dislocation); the potential discrepancy on NGF origin; specificity of NGF antibodies used in the current study. Some important information pertaining to ethics and welfare is required. Human heart samples were obtained from an archived biobank. The authors cite a national committee of Bioethics and reference the year 2006. Did the committee provide approval for the use of the samples in this study, or does the citation reference a blanket approval for such work? There is no statement on the issue of consent. If these are historical samples, or given that they are clinical samples obtained in the course of routine practice (autopsy), it may be that the committee has waived this requirement. Please provide clarity on this matter. Please provide the approval code/number for the study in rats. Top of page 8: Please change "sacrificed" to killed or euthanised. Rats were killed by cervical dislocation. For rodents greater than 150g body mass, it is a requirement that animals are first sedated. Were adult rats sedated prior to cervical dislocation? Please provide details.

Senior Editor Comments:

I concur with the reviewing editor.

Au: We thank the Editors and reviewers for the thorough and attentive revision of our manuscript, and we appreciate that, thanks to the suggestions upon initial evaluation, our data strengthen the concept of retrograde signaling from sympathetic neurons to cardiomyocytes in the heart. We understand the reviewers' concerns, which are addressed in the 'point-by-point' response below.

'Point-by-point' response to reviewers:

Referee #1:

In this revised manuscript by Dokshokova, Franzoso et al, the authors have addressed my questions. I have no additional comments.

Au: We are delighted that our reviews met the referee's requests.

Referee #2:

This is much improved. Two further comments for clarification.

1. A further comment in the limitations is also required given the NGF data in neurons. In Figure 4C NGF expression appears in the neuron axon outside of the cardiomyocyte, which would imply that NGF is expressed in the varicosity, and not in the cardiomyocyte. Other studies have suggested that NGF is released by the vasculature, and that NGF is not expressed in the cardiomyocyte. How do the authors explain this discrepancy? If the authors believe that NGF is in the neurons, then please provide a western blot or acknowledge the limitation here.

Au: In this work we are demonstrating the CMs sustain the innervating neurons by providing NGF at the level of the neurocardiac junction. NGF signal inside the neuronal varicosities is thus expected, as NGF from CMs binds to TrkA in the membrane of the neuronal varicosity and is internalized by the neuron and transported back to the neuronal soma where it activates pro-survival signaling pathways. *In vitro* data in Figure 4C are in line with *in vivo* evidence described in paragraph 1 of the Result section. We apologize with the reviewer if the text made him/her misunderstand this concept.

“Staining of the sections with an anti-NGF antibody showed immunoreactive vesicles in CMs, which clustered in a roughly 2µm deep submembrane space, and appeared to mirror the position of the pre-synaptic neuronal varicosities, in the majority (>60%) of neuro-cardiac interfaces analysed (**Fig.1B**). Consistently, sub-microscopic analysis of the immunostained sections revealed high intensity NGF puncta (in green) spread in the CM cytosol, accompanied by neurotrophin clusters aligned along the portion of the CM membrane directly contacted by the SN (**Fig.1C**). This was in line with the results of morphometric image analysis aimed to trace pre-synaptic SNAP-25 and post-synaptic NGF distribution, showing that the highest NGF fluorescence intensity, in the CM, corresponded to the peak of SNAP-25 signal on the neuron (**Fig.1D**). In addition, NGF was detected along the neuronal process, suggesting that CM-released neurotrophin could be sensed and locally internalized, *via* TrkA, by the contacting SN (**Fig.1B**). Interestingly, the presence of NGF in CM-innervating neuronal processes and the preferential localization of the neurotrophin in the CM submembrane space, at neuro-cardiac contacts, were detected, and confirmed by morphometric analysis, in myocardial samples from post-mortem human hearts, proving the principle that the aspects described above hold true in the normal human heart (**Fig.2A-C**).

Thus, the arrangement of NGF vesicles in CMs and of NGF-sensing receptors on SNs, suggests that exchange of neurotrophin between the two cell types may take place locally, and that the NCJ may thus be poised to sustain retrograde (myocyte-to-neuron) neurotrophic signaling.”

Concerning the second point, we are aware that also vascular cells express NGF in addition to other neurotrophins (i.e. NT-3) and that they are key in postnatal heart innervation. However, a large body of evidence has clearly shown that NGF is synthesized by CMs, whose secretion is affected in several disease conditions (*Rana et al JMCC 2009; Meloni Circ res 2010; Habecker et al., J Physiol 2016, to name a few*). We thus do not believe that this is a discrepancy that needs to be commented on.

2. Why has the western blot band for NGF in the adult heart been cut, and what was the band above NGF, which has been cropped off in adult heart sample 1. Does this mean the NGF antibody also bound to other, non-specific targets. Please clarify?

Au. We can exclude that the band at approximately 19 kDa in WB of Fig.4 panel B is due to a non-specific binding of the antibody to other neurotrophins/non-specific targets for several reasons: i) other neurotrophins expressed by the heart have different molecular weight; ii) specificity of our anti-NGF antibody was tested, before starting experiments, by IF and WB analyses in NGF-expressing vs. NGF downregulated cells, as well as with pure NGF; iii) no additional bands were detected in extracts from culture cells expressing in addition to NGF other neurotrophins. Based on the literature, the 19kDa band in extracts from adult heart presumably refers to post-translational protein modification (i.e. glycosylation) which has been demonstrated to not affect NGF biological activity (Murphy et al., J Biol Chem 1989). In any case to compare cardiac NGF content in neonatal vs. adult hearts we took into consideration the band of mature/unmodified NGF.

Referee #3 (ethics review):

Thank you for submitting your manuscript to The Journal of Physiology. Some important information pertaining to ethics and welfare is required.

Au: We thank the Ethics reviewer for noticing some inconsistencies in the text which has been amended as indicated.

1) Human heart samples were obtained from an archived biobank. The authors cite a national committee of Bioethics and reference the year 2006. Did the committee provide approval for the use of the samples in this study, or does the citation reference a blanket approval for such work? There is no statement on the issue of consent. If these are historical samples, or given that they are clinical samples obtained in the course of routine

practice (autopsy), it may be that the committee has waived this requirement. Please provide clarity on this matter.

As it concerns the use of biopsies from post-mortem heart samples, use is regulated by the directives on the use of biological samples in research. Obtainment of informed consent is waived as i) samples were anonymized to investigators and ii) it was impossible to receive consent, given that the biopsies are obtained from residual material pathologic assessments performed in post-mortem examinations. The principles underlying these rules are detailed in general and specific terms in the “Recommendation (CM/Rec(2016)6) of the Committee of Ministers to member States on research on biological materials of human origin”, released by the Council of Europe, in accordance with the Convention for the Protection of Human Rights and Fundamental Freedoms (ETS No. 5) and the Convention on Human Rights and Biomedicine (ETS No. 164) and of its Additional Protocol concerning biomedical research (CETS No. 195). In practice, such Recommendation represents a *blanket* approval for use of such biologic material.

These considerations explain why the summarizing recommendation, updated in the revised manuscript version with the most recent version, dated 2016, has been cited in the methods section.

2) Please provide the approval code/number for the study in rats.

Au: Code for study in rats (VIMM C-53 and C-54) are indicated in the method section. Code numbering refers to studies approved by the institutional ethical committee on animal experiments which was the sole legal requirement for performing research, when experiments were performed.

3) Top of page 8: Please change "sacrificed" to killed or euthanised.

Au: Amended accordingly.

4) Rats were killed by cervical dislocation. For rodents greater than 150g body mass, it is a requirement that animals are first sedated. Were adult rats sedated prior to cervical dislocation? Please provide details.

Au: Details have been added to the relative method section.

“Origin and Source of Animals. In this study, we used P1-P3 and adult (3 mo.) Sprague-Dawley male rats (Harlan, Milan, Italy). Animals were maintained in individually ventilated cages in an Authorized Animal Facility (authorization number 175/2002A) under a 12:12 hours light/dark cycle at a controlled temperature and had access to water and food available *ad libitum*. Rats were killed by cervical dislocation (in accordance with Annex IV of European Directive 2010/63/EU). In adult rats, sedation with 3% isoflurane (v:v in O₂) was performed before cervical dislocation.

Dear Dr Mongillo,

Re: JP-RP-2022-282828R2 "Nerve Growth Factor transfer from cardiomyocytes to innervating sympathetic neurons activates TrkA receptors at the neuro-cardiac junction" by Marco Mongillo, Lolita Dokshokova, Mauro Franzoso, Anna Di Bona, Nicola Moro, Jose L. Sanchez-Alonso, Valentina Prando, Michele Sandre, Cristina Basso, G Faggian, Hugues Abriel, Oriano Marin, Julia Gorelik, and Tania Zaglia

I am pleased to tell you that your paper has been accepted for publication in The Journal of Physiology.

NEW POLICY: In order to improve the transparency of its peer review process The Journal of Physiology publishes online as supporting information the peer review history of all articles accepted for publication. Readers will have access to decision letters, including all Editors' comments and referee reports, for each version of the manuscript and any author responses to peer review comments. Referees can decide whether or not they wish to be named on the peer review history document.

The last Word version of the paper submitted will be used by the Production Editors to prepare your proof. When this is ready you will receive an email containing a link to Wiley's Online Proofing System. The proof should be checked and corrected as quickly as possible.

Authors should note that it is too late at this point to offer corrections prior to proofing. The accepted version will be published online, ahead of the copy edited and typeset version being made available. Major corrections at proof stage, such as changes to figures, will be referred to the Reviewing Editor for approval before they can be incorporated. Only minor changes, such as to style and consistency, should be made a proof stage. Changes that need to be made after proof stage will usually require a formal correction notice.

All queries at proof stage should be sent to TJP@wiley.com

Are you on Twitter? Once your paper is online, why not share your achievement with your followers. Please tag The Journal (@jphysiol) in any tweets and we will share your accepted paper with our 23,000+ followers!

Yours sincerely,

Bjorn Knollmann
Senior Editor
The Journal of Physiology

P.S. - You can help your research get the attention it deserves! Check out Wiley's free Promotion Guide for best-practice recommendations for promoting your work at www.wileyauthors.com/eoo/guide. And learn more about Wiley Editing Services which offers professional video, design, and writing services to create shareable video abstracts, infographics, conference posters, lay summaries, and research news stories for your research at www.wileyauthors.com/eoo/promotion.

*** IMPORTANT NOTICE ABOUT OPEN ACCESS ***

Information about Open Access policies can be found here <https://physoc.onlinelibrary.wiley.com/hub/access-policies>

To assist authors whose funding agencies mandate public access to published research findings sooner than 12 months after publication The Journal of Physiology allows authors to pay an open access (OA) fee to have their papers made freely available immediately on publication.

You will receive an email from Wiley with details on how to register or log-in to Wiley Authors Services where you will be able to place an OnlineOpen order.

You can check if your funder or institution has a Wiley Open Access Account here <https://authorservices.wiley.com/author-resources/Journal-Authors/licensing-and-open-access/open-access/author-compliance-tool.html>

Your article will be made Open Access upon publication, or as soon as payment is received.

If you wish to put your paper on an OA website such as PMC or UKPMC or your institutional repository within 12 months of publication you must pay the open access fee, which covers the cost of publication.

OnlineOpen articles are deposited in PubMed Central (PMC) and PMC mirror sites. Authors of OnlineOpen articles are permitted to post the final, published PDF of their article on a website, institutional repository, or other free public server, immediately on publication.

Note to NIH-funded authors: The Journal of Physiology is published on PMC 12 months after publication, NIH-funded authors DO NOT NEED to pay to publish and DO NOT NEED to post their accepted papers on PMC.

EDITOR COMMENTS

Reviewing Editor:

Thank you for the revision. All concerns from the reviewers have been addressed.

Senior Editor:

Excellent and impactful work!

REFEREE COMMENTS

Referee #2:

Thank you for clarifying my points. This is a nice study.

Referee #3:

Thank you for addressing the points raised. There are no further issues.

2nd Confidential Review

18-Mar-2022